# Sea ice export through the Fram Strait derived from a combined model and satellite data set

Chao Min[1,2,3], Longjiang Mu[4], Qinghua Yang[1,2,3], Robert Ricker[4], Qian Shi[1,3], Bo Han[1,3], Renhao Wu[1,3], Jiping Liu[5]

[1]School of Atmospheric Sciences and Guangdong Province Key Laboratory for Climate Change and Natural Disaster Studies, Sun Yat-sen University, Zhuhai, 519082, China
[2]State Key Laboratory of Numerical Modeling for Atmospheric Sciences and Geophysical Fluid Dynamics, Institute of Atmospheric Physics, Chinese Academy of Sciences, Beijing, 100029, China
[3]Southern Marine Science and Engineering Guangdong Laboratory (Zhuhai), Zhuhai, 519082, China
[4]Alfred Wegener Institute, Helmholtz Centre for Polar and Marine Research, Bremerhaven, 27570, Germany
[5] Department of Atmospheric and Environmental Sciences, University at Albany, State University of New York, New York, 12222, US

*Correspondence to*: Longjiang Mu (longjiang.mu@awi.de) and Qinghua Yang (yangqh25@mail.sysu.edu.cn)

**Abstract.** Sea ice volume export through the Fram Strait plays an important role on the Arctic freshwater and energy redistribution. The combined model and satellite sea ice thickness (CMST) data set assimilates CryoSat-2 and Soil Moisture and Ocean Salinity (SMOS) thickness products together with satellite sea ice concentration. The CMST data set closes the gap of stand-alone satellite-derived sea ice thickness in summer, and therefore allows us to estimate sea ice volume export during the melt season. In this study, we first validate the CMST data set using field observations, and then estimate the continuous seasonal and interannual variations of Arctic sea ice volume flux through the Fram Strait from September 2010 to December 2016. The results show that seasonal and interannual sea ice volume export vary from about -240 (±40) to -970 (±60) km$^3$ and -1970 (±290) to -2490 (±280) km$^3$, respectively. The sea ice volume export reaches its maximum in spring and about one third of the yearly total volume export occurs in the melt season. The minimum monthly sea ice export is -11 km$^3$ in August 2015 and the maximum (-442 km$^3$) appears in March 2011. The seasonal relative frequencies of sea ice thickness and drift suggest that the Fram Strait outlet in summer is dominated by sea ice that is thicker than 2 m and drifting with relatively slow seasonal mean speed of about 3 km d$^{-1}$.

## 1 Introduction

The sea ice extent and volume in the Arctic region undergo a decline for the past decades and will likely continue to decrease (Comiso and Hall, 2014; Meier et al., 2014; Stroeve and Notz, 2015). The decline of ice extent changes the surface albedo, and as a consequence, the absorption of solar shortwave radiation increases. The variability of ice volume, however, exerts influence on heat, freshwater budget and weather systems in the lower latitudes (Gregory et al., 2002; Tilling et al., 2015). Correspondingly, both the thermodynamic processes and dynamic processes can affect Arctic sea ice mass budget (Ricker et al, 2018). The sea ice outflow driven by atmospheric circulation is an important component of dynamic processes. The Fram

Strait serves as the primary outlet of the Arctic sea ice export (Krumpen et al., 2016). Moreover, the ice outflow through the strait into the Nordic Seas covers approximately 25% of the total Arctic freshwater export (Lique et al., 2009; Serreze et al., 2006).

Variations of satellite-based Arctic sea ice volume and sea ice export through the Fram Strait have been estimated by numerous studies (Bi et al., 2018; Kwok and Cunningham, 2015; Ricker et al., 2018; Spreen et al., 2009). Nevertheless, in terms of the volume flux, the primary focus of these studies are the variations during the freezing season (October-April). This is due to the limitations in retrieving sea ice thickness and motion by satellite remote sensing during the melt season (May-September). It is mainly caused by more melt ponds and statured water vapor in the sea ice surface, which restrains satellite-based ice thickness limited to the cold season only (Mu et al., 2018a). The speed-up of sea ice drift usually accompanies with thin sea ice, meanwhile the faster sea ice drift the larger retrieving errors there would be (Spreen et al., 2011; Sumata et al., 2014). Melting sea ice with a less scattering surface could significantly suppress the signal-to-noise ratio and obstruct the employment of satellite imagery to retrieve ice drift. For above-mentioned reasons, the spaceborne sea ice drift data usually induce more uncertainties in the melt season. All these deficiencies make the estimate of the Arctic sea ice thickness and drift variations all year round difficult with only satellite sea ice data.

Sea ice volume flux, compared to area flux, could reflect the sea ice mass balance in a more comprehensive way. However, the amounts of Fram strait sea ice volume export during the freezing season do not demonstrate a conspicuous growth or decline trend (Ricker et al., 2018; Spreen et al., 2009). And the variation of the melt season ice volume flux through the Fram Strait still remains a query owing to the fact that sea ice thickness observations are sparse in the melt season, and so does the yearly total amount of ice volume flux. In terms of sea ice volume flux, Ricker et al. (2018), Bi et al. (2018) and Zamani et al. (2019) point out that the variation of ice drift plays the major role in determining the annual and interannual ice volume export variability. Due to thermodynamic growth and deformation, sea ice thickness on the other hand drives the increase in the seasonal cycle of the exported volume. For this reason, an accurate data set of sea ice drift and thickness is crucial to better estimate sea ice volume output.

Employing the benefits of both the CryoSat-2 (CS2) and the Soil Moisture and ocean Salinity satellite (SMOS) sea ice thickness products, the new data set (combined model and satellite thickness, CMST) that assimilates these data together with satellite-derived sea ice concentration (Mu et al., 2018a; Mu et al., 2018b) provides the daily sea ice thickness, concentration and drift estimates simultaneously. Moreover, taking advantages of model dynamics and sea ice concentration assimilation, the new sea ice data set extends to cover the melt season when satellite thickness data are limited (Mu et al., 2018a). Previous results reveal that CMST data even have some advantages among the statistically merged satellite data CS2SMOS and Pan-Arctic Ice-Ocean Modelling and Assimilation System (PIOMAS) thickness product when comparing with the in-situ observations (Mu et al., 2018a). Therefore, the CMST sea ice product enables us to examine the all-year-round changes in sea ice volume export through the Fram Strait for 2010-2016, during a time when Arctic sea ice is undergoing dramatic changes. Further, we also calculate the sea ice thickness, concentration and drift frequency distributions along the main sea ice export gate all-year-round.

This paper is organized as follows. Section 2 describes the data used to validate the CMST data set and the method used to derive the volume flux. In section 3, firstly, we evaluate the performance of CMST data. Then, we estimate the continuous seasonal and interannual variation of sea ice thickness, concentration and drift in the Fram Strait. Also, the all-year-round variability of sea ice volume export though the Fram Strait is calculated. Uncertainty in our volume flux estimate is discussed in Section 4. Concluding remarks are given in Section 5.

## 2 Data and Methods

### 2.1 CMST sea ice data

The CMST sea ice data in addition to ice thickness and concentration also provide the modelled ice drift. They are generated by an Arctic reginal ice-ocean model accompanying with CS2, SMOS sea ice thickness and SSMIS sea ice concentration assimilation. This Arctic regional model (Losch et al., 2010; Mu et al., 2017; Nguyen et al., 2011; Yang et al., 2014) is configured on the basis of the Massachusetts Institute of Technology generation circulation model (MITgcm) (Marshall et al., 1997). To reflect the impacts of atmospheric uncertainties on the sea ice data assimilation, the atmospheric ensemble forecasts of the United Kingdom Met Office (UKMO) Ensemble Prediction System (EPS; https://www.ecmwf.int/en/research/projects/tigge) are used as atmospheric forcing (Mu et al., 2018b; Yang et al., 2015; Yang et al., 2016). The Parallel Data Assimilation Framework (PDAF, Nerger and Hiller, 2013; http://pdaf.awi.de) is applied to assimilate satellite thickness (e.g., SMOS thickness data thinner than 1 m and weekly mean CS2 thickness data) and concentration data (provided by the Integrated Climate Data Center, http://icdc.cen.uni-hamburg.de). More details about this assimilation process can be found in previous studies (Mu et al., 2018a; Mu et al., 2018b). CMST provides grid cell-averaged ice thickness, i.e., the effective ice thickness (Mu et al., 2018a; Schweiger et al., 2011) with a resolution about 18 km. Further taking advantage of model dynamics and ice concentration assimilation, the daily CMST thickness data in summer are also available from September 2010 to December 2016. Although the CMST data do not span the recent two years (i.e., year of 2017 and 2018), it does cover the year of the lowest sea ice extent record at that time (i.e., 2012 and 2016) (Parkinson and Comiso, 2013; Petty et al., 2018).

### 2.2 OSI SAF drift data

As suggested by Sumata et al. (2014), the merged OSI SAF sea ice drift product (OSI-405) reveals a better performance than other low-resolution sea ice drift products in the Fram Strait. Thus, we use it for comparison with CMST drift data when calculating sea ice volume export. The merged drift data can be download from the Ocean and Sea Ice Satellite Application Facility (OSI SAF, http://www.osi-saf.org/?q=content/sea-ice-products). The merged drift products are retrieved from multiple sensors and channels (shown in Table 1) in order to supplement data gaps in the single-sensor products. A more detailed description can be seen in the Low Resolution Sea ice Drift Product User's Manual (http://osisaf.met.no/p/ice/lr_ice_drift.html).

### 2.3 NSIDC sea ice drift

The latest released Polar Pathfinder Daily 25 km EASE-Grid sea ice drift data from the National Snow and Ice Data Center (NSIDC, https://nsidc.org/data/nsidc-0116/versions/4) are also used to evaluate the CMST drift. These data cover both the melt season and the freezing season and widely used in the modeling and data assimilation (Miller et al., 2006; Stark et al., 2008). The input sea ice motion data sets are obtained from AVHRR, AMSR-E, SMMR, SSM/I, SSM/I, International Arctic Buoy Program (IABP) buoys and National Center for Environmental Prediction (NCEP) / National Center for Atmospheric Research (NCAR) Reanalysis wind data. More descriptions can be seen in the NSIDC ice motion user guide (https://nsidc.org/data/nsidc-0116/versions/4).

### 2.4 Sentinel-1 SAR sea ice drift

To further validate the CMST sea ice drift in the Fram Strait, the sea ice drift data retrieved from Sentinel-1 Synthetic Aperture Radar (SAR) images are used as the reference products (https://www.ncbi.nlm.nih.gov/pmc/articles/PMC5999601/). Based on the different polarization channels, thousands of HH and HV polarization images are calculated as monthly mean sea ice drift at 79°N along the gate from 15°W to 5°E (Muckenhuber et al., 2018). These SAR drift data are derived by an open-source feature-tracking algorithm (Muckenhuber et al., 2016). Owing to the better performance of the HV polarization channel (Muckenhuber et al., 2016), we only use the southward velocity component of HV polarization for the validation. More information about the Sentinel-1 SAR sea ice drift can be obtained in the previous studies (Muckenhuber et al., 2016; Muckenhuber et al., 2018).

### 2.5 HEM sea ice thickness

For the purpose of evaluating the performance of CMST sea ice thickness, the helicopter-borne electromagnetic induction sounding (HEM) sea ice thickness (https://data.npolar.no/dataset/1ed8c57e-8041-42fd-95bb-cfe4e181e9b8) is utilized for intercomparision. This HEM measurement campaigns consist of 9 separate flights implemented in the Fram Strait from August to September, 2014. The helicopter-measured sea ice thickness is named as "total thickness" including snow layer. Thus, following Krumpen et al. (2016), we assume the thickness of snow or weathered ice is 0.1 m, i.e., we subtract the 0.1 m snow thickness from the "total thickness" in the later calculation. Sea ice concentration is low in the operational areas during this period and the data have not been adjusted with sea ice concentration. Because the CMST model thickness are effective thickness (i.e., mean thickness over the model grid), for easy comparison, and as recommended by the data providers (https://data.npolar.no/dataset/1ed8c57e-8041-42fd-95bb-cfe4e181e9b8), we adjust this data with the CMST ice concentration to obtain daily mean ice floe thickness.

## 2.6 ULS sea ice thickness

The upward looking sonars (ULS) measurement (moored at 79°N, 5°W) in the Fram Strait is deployed and maintained by the Norwegian Polar Institute. Since ULS measures sea ice draft, the derived sea ice thickness is less affected by uncertainties in the snow layer depth and ice density. Moreover, the ULS provides year-round measurements and are therefore used to validate the CMST thickness. More details about the ULS data can be found in previous work (Hansen et al., 2013). In this study, we use a 1-year data set of monthly mean sea ice thickness from September, 2010 to August, 2011.

## 2.7 Retrieving methods in sea ice volume export

The sea ice thickness, concentration and drift in CMST data set are provided on a cube-sphere grid with a resolution of 18 km. Both sea ice variables in CMST and the OSI-405 merged data are projected to the geographic coordinates at first. Following Krumpen et al. (2016) and Ricker et al. (2018), we define the Fram Strait export gate with zonal and meridional components as shown in Figure 1. The zonal gate is situated at 82°N between 12°W and 20°E, and the meridional gate is located at 20°E between 80.5°N and 82°N. The chosen gates are dedicated to decrease errors and bias in low resolution drift data and thickness data from satellite (Krumpen et al., 2016; Ricker et al., 2018). Secondly, we use linear interpolation method to interpolate the CMST data and OSI SAF data onto the zonal gate with a spatial resolution of 1° and onto the meridional gate with a spatial resolution of 0.15°, which is of the purpose to better match the model grids with the interpolated grids.

Following Ricker et al. (2018), we also define the negative values represent ice volume loss from the Arctic Basin through the outlet and the sea ice volume flux can be estimated as following formulas:

$$Q_x = L_x \, H_x \, v, \tag{1}$$

$$Q_y = L_y \, H_y \, u, \tag{2}$$

where $L_x$ is the size of zonal interpolated grid and $L_y$ is the size of meridional interpolated grid. $H_x$ and $v$ are the interpolated effective ice thickness and meridional velocity at the zonal gate. $H_y$ and $u$ are the interpolated effective ice thickness and zonal velocity at the meridional gate. Note that ice concentration information is not involved in equations (1) and (2) because the calculation process of CMST model effective ice thickness has already taken ice concentration information into account.

The total sea ice volume export ($Q_{EX}$) through the Fram Strait is obtained by adding the zonal ice volume flux ($Q_x$) and meridional ice flux ($Q_y$) together:

$$Q_{EX} = Q_x + Q_y, \tag{3}$$

Uncertainties of sea ice volume export ($\delta_{Q_x}$) are evaluated as:

$$\delta_{Q_x} = L \sqrt{(H \, \delta_v)^2 + (v \, \delta_H)^2}, \tag{4}$$

This strategy is used to estimate the expected uncertainties of volume flux via the zonal gate. $\delta v$ and $\delta_H$ represent ice drift
uncertainty and ice thickness uncertainties, respectively. Expected sea ice volume flux uncertainties along the meridional gate
can be determined by the similar method of (4).

Sea ice volume export derived from CMST thickness and drift is represented by M2 in detail in Table 2 (Section 3.2). The
results derived from CS2 thickness and OSI SAF drift for Ricker et al. (2018) are represented by R. To investigate the flux
biases due to the existing deviations between the CMST and the CS2 thickness data, CMST thickness and OSI SAF drift are
also used to calculate the flux that is shown by M1.

## 3 Results

### 3.1 Validation of CMST data

Firstly, the field and satellite-based observations are used to evaluate the performance of CMST sea ice data in the Fram Strait.
The mean sea ice drift and thickness of nearly 6 years' CMST data are shown in Figure 1a. The mean sea ice thickness is
distributed as expected (Tilling et al., 2015; Kwok et al., 2018), e.g., the relatively thicker ice, which is more than 2.5 m,
mainly distributes in the north of Greenland and the Canadian Arctic Archipelago and the sea ice becomes thinner towards the
Eurasia coasts (Figure 1a). We then compare the mean difference between the CMST drift and the latest released sea ice drift
data (V4) from the NSIDC. The circulation patterns (the Transport Drift and the Beaufort Gyre) and magnitudes distributions
of the two sea ice drift data (CMST vs. NSIDC) are quite similar (Figure not shown). The relatively larger differences of sea
ice drift speed are found along the coast of Greenland and ice edge, which is shown in Figure 1b. It is noticeable that the mean
sea ice drift speed of CMST is larger than the NSIDC in most areas. This may suggest that the CMST sea ice drift performs
better than NSIDC drift data in the Fram Strait for that NSIDC drift data usually exist underestimations in sea ice velocity
(Sumata et al., 2015; Sumata et al., 2014). For further validation of CMST sea ice velocity, we compare the CMST southward
velocities that affect sea ice volume flux most with high-resolution Sentinel-1 SAR sea ice drift data. Results (Figure 1c and
1d) show that both CMST drift and NSIDC drift generally overestimate the southward velocities near the Greenland but
underestimate the velocity far away from the Greenland. Nevertheless, monthly mean CMST drift data show a better
performance than NSIDC drift data especially near the Greenland.

Further assessments of CMST thickness and drift data are shown in Figure 2. The geography map (Figure 2a) shows the
trajectories of HEM measurement campaigns and the site of ULS. Helicopter-borne daily mean sea-ice thickness is used to
evaluate the CMST thickness data in the Fram Strait. Monthly CMST sea ice thickness is also compared with the thickness
derived from the ULS data (shown in Figure 2c). Note that the comparison period for CMST thickness and ULS thickness is
from September 2010 to August 2011, since the ULS data afterwards have not been available for this study. Monthly mean
CMST sea ice drift over the entire Fram Strait gate is evaluated with OSI SAF drift used in Ricker et al. (2018) within the
same period from September 2010 to December 2016 and the same domain defined before. The correlation coefficient (CC),

the relative bias (RB) and the root-mean-squared error (RMSE) are explored to quantify the comparison. These statistic metrics are calculated as follows (Chen et al., 2013; Zhang et al., 2019):

$$CC = \frac{Cov(CMST, OBS)}{\sigma_{CMST}\,\sigma_{OBS}},$$ (5)

$$RB = \frac{\sum(CMST - OBS)}{\sum OBS},$$ (6)

$$RMSE = \sqrt{\frac{(CMST - OBS)^2}{N}},$$ (7)

where Cov represents the covariance operator, $\sigma$ is the standard deviation, and the number of the observations (OBS) is indicated by N.

Statistical analysis between CMST and HEM sea ice thickness shows that the CC, RB and RMSE are 0.59, 15.13% and 0.66 m, respectively. Furthermore, statistics indicate that the CMST data is comparable to ULS measurements with a CC of 0.68, a low RB (1.74%) and RMSE (0.328 m). Note that the CMST thickness has been already quantitatively evaluated with more

observation records by a previous study (Mu et al., 2018a) and exhibits some advantages over the widely used CS2SMOS and PIOMAS thickness data. The CC between CMST drift and OSI SAF drift shows a high correlation of 0.93 (Figure 2d) in the freezing season (October-April). The RB (-6.05%) and RMSE (0.985 km d[-1]) are also relatively quite low. These statistical metrics suggest a good performance of CMST over the Fram Strait outlet in reproducing the real sea ice drift and thickness.

## 3.2 Sea ice thickness, concentration and drift variation

In this study, the spring, summer, autumn and winter span from March to May, June to August, September to November and December to February, respectively. The all-year-round seasonal variation of Arctic sea ice thickness and concentration are shown in Figures 3 and 4. During the study period, both the Arctic sea ice thickness and concentration show a significant seasonal variation, e.g., the sea ice thickness reach its maximum in spring (except for 2013), while the sea ice concentration has a peak value in spring/winter.

As shown in Figure 3, the distribution of sea ice thickness along the Fram Strait zonal gate features thicker sea ice in the east of Greenland than that in the west of Svalbard, showing a gradually thinning trend from west to east. And along with the meridional gate, sea ice is thickening from the northern Svalbard to the central Arctic Ocean, which is in line with other studies (Hansen et al., 2013; Kwok et al., 2004; Krumpen et al., 2016; Vinje et al., 1998). Note that the sea ice thickness hits its minimum in the autumn of 2011, and such anomaly is also found in previous studies based on sea ice satellite data (Kwok and

Cunningham, 2015; Tilling et al., 2015). Also, it is notable that the mean thickness of the winter 2013 arises a significant thickening comparing with other winters. This remarkable thickening may be linked to the anomalously cooling in 2013 which enhances the thermodynamic ice growth (Tilling et al., 2015).

Further analysis on the sea ice volume in the Arctic basin shows a typical seasonal variation with the minimum in autumn and the maximum in spring. The Arctic sea ice volume undergoes a minimum season in the autumn of 2011 ($6.93\times10^3$ km$^3$) and

reaches a maximum of $20.19 \times 10^3$ km$^3$ in the spring of 2014. A maximum (minimum) sea ice extent does not correspond to a maximum (minimum) volume. For instance, the sea ice extent minimum ($5.17 \times 10^6$ km$^2$) and maximum ($10.87 \times 10^6$ km$^2$) are each found in autumn of 2012 and in spring of 2013, while the sea ice volume minimum ($6.93 \times 10^3$ km$^3$) happens in autumn of 2011 and the maximum of $20.19 \times 10^3$ km$^3$ occurs in spring of 2014. The trends of the temporal variation of Arctic ice volume and extent are similar to the results from Tilling et al. (2015) and Kwok and Cunningham (2018).

The sea ice thickness, concentration and drift averaged over the entire Fram Strait gate are shown in Figures 5, 6 and 7, respectively. We also compare these sea ice variables with Ricker at al. (2018). The results show that the CMST ice thickness and drift are smaller than that of CS2 and OSI SAF, while the CMST ice concentration is a little larger than OSI SAF ice concentration. Such thinner CMST sea ice thickness found in the Fram Strait is discussed to be reasonable in Mu et al. (2018b) because of the assimilation of SMOS thickness data. The previous study shows that the mean Arctic-wide OSI SAF drift is

slightly larger than IABP/D buoy ice drift (Sumata et al., 2014), which suggests the slight underestimation of CMST drift seems also tenable. Further validation with more ice drift data over the Arctic basin (e.g., buoy drift data) is needed; however, it is beyond the scope of this work. In terms of variation trend, they are in good agreement with those of Ricker at al. (2018). As shown in Figures 5 and 7, the averaged sea ice thickness and drift reveal a significant seasonal cycle. That is, the variations of sea ice thickness and motion always accompany with spring increase and autumn decrease. The analysis of ice concentration

shows a steadily low values in the melt season. And the 6-year mean sea ice thickness, concentration and drift averaged over the entire Fram Strait gate are about 1.7 m, 85% and 5 km d$^{-1}$.

Following Ricker et al. (2018), the relative standard deviation (RSD=SD/mean) is used to measure the effects of different sea ice variables on the variability of the ice volume output. Variables with a lager RSD contribute to a greater impact on the volume variation. As shown in Figures 5, 6 and 7, the RSD of ice thickness is 0.30 which is about twice of ice concentration

(0.14). The ice drift is the largest contributor with an RSD of 0.50. It is shown that the ice drift with maximal RSD is more likely to affect variations in sea ice volume flux, which is corresponding to the previous findings in Kwok et al. (1999), Ricker et al. (2018) and Bi et al. (2018).

To analyze the respective contributions of ice drift and ice thickness to the seasonal variation of sea ice export, the frequency distributions of seasonal sea ice thickness (Figure 8), drift (Figure 9) and concentration (not shown owing to the minimum

RSD) along the Fram Strait outlet are further calculated. Specifically, we define the relative frequency (RF) as following:

$$RF = \frac{n}{N_{grids}}, \tag{8}$$

where $n$ represents the number of the grid cells accounted in different thickness bins, and $N_{grids}$ is the sum of $n$. As suggested by Figure 8, the thickness along the zonal gate is much thicker than the meridional gate. Thin ice is more observed in autumn over the zonal gate according to the RF distribution in Figure 8. Although the maximum thickness over the entire Fram Strait

occurs in May and June (Figure 5), higher RF in thick ice bins are found in summer (June, July and August in our definition) over zonal gate. Over the meridional gate, the ice thickness in summer and spring is almost uniformly distributed, while in

autumn and winter, high RFs are more found in thin ice bins. In statistics, the seasonal mean sea ice thicknesses are 2.06 m for spring, 2.11 m for summer, 1.32 m for autumn and 1.43 m for winter over the entire outlet, respectively. Nevertheless, the mean relative frequency of sea ice drift distribution (Figure 9) shows that the ratio of summer sea ice drift lower than 6 km d$^{-1}$ is in the majority (more than 90% of zonal gate) indicating that the sea ice drift is much slower than other seasons. Also, the ice drift along the zonal gate is usually faster than the meridional gate and the meridional sea ice velocities are slower than 6 km d$^{-1}$ during summer. The seasonal mean sea ice velocity over the entire gate is larger than 5 km d$^{-1}$ except that is 3 km d$^{-1}$ in summer. And it can be found that the spring and winter ice concentration along the zonal gate is larger than that of summer and autumn.

## 3.3 Sea ice volume export through the Fram Strait

In this section, sea ice volume export over all seasons is investigated. Firstly, the examination of monthly Arctic sea ice volume export through the Fram Strait is shown in Table 2. Both our results and Ricker et al. (2018) find that the maximum monthly sea ice export takes place in March 2011. The maximum of CMST data is -442 km$^3$ that is less than that (-540 km$^3$) of Ricker et al. (2018). Consistently, the lowest sea ice output for each study occurs in February 2011 when excluding the melt season (May-September). The minimum of the results shown in Ricker et al. (2018) is -21 km$^3$ while that is -34 km$^3$ in CMST data. Although there are some differences in flux calculated based on CMST data and CryoSat-2 thickness and OSISAF drift data, both the estimations show a similar trend in annual cycle. Furthermore, the CMST data can provide sea ice variables (e.g., sea ice thickness, concentration and drift) in the melt season that remote sensing retrieval data cannot cover. Taking advantage of CMST data, this study is trying to fill the research gap in the summer sea ice volume export. It is found that another minimum of ice export occurs in August 2015 because of the rather slow mean sea ice velocity (shown in Figure 11) during the study period. The minimum value for CMST is -11 km$^3$ that is 10 km$^3$ less than -21 km$^3$ (R) in February 2011 and 23 km$^3$ less than that for M2.

Moreover, the seasonal variation of sea ice export though Fram Strait is shown in Figure 10. The ice volume output shows a significant seasonal variation. The seasonal maximums are found in spring of all years (2011-2016) and the low values usually occur in summer and autumn. The maximum seasonal ice export of -970 (±60) km$^3$ (sea ice volume export has been rounded off to significant figures in seasonal and interannual time scales) takes place in the spring of 2012 owing to both simultaneously faster ice drift and thicker ice thickness, while the minimum flux of -240 (±40) km$^3$ occurs in autumn of 2016 caused by simultaneously rather slower ice motion and thinner ice thickness. Unlike in other autumns, the ice volume export of autumn 2013 abnormally increases and reaches -620 (±60) km$^3$. This abnormal increase can be also explained by the faster ice drift (shown in Figure 9).

Furthermore, we standardize the sea ice volume export, ice drift and thickness and then calculate the correlations of determination ($R^2$) between monthly sea ice volume export and thickness, and also for sea ice drift (shown in Figure 11). For monthly mean sea ice flux and drift, $R^2$ is 0.77, which is much higher than that against thickness (0.16). This result shows that the sea ice drift variation contributes more to sea ice flux variation on its monthly variability. However, when averaged over

seasonal time scale, both the sea ice drift and thickness become significant factors for their close $R^2$ within the range of 0.36-0.46. Analogously, this conclusion was pointed out by Ricker et al. (2018) and Bi et al. (2016). In addition, the Arctic Oscillation (AO) and North Atlantic Oscillation (NAO) index are used to analyze the possible links between atmospheric circulation and sea ice volume flux through the Fram Strait (Figure 12). The AO and NAO indexes are both obtained from National Oceanic and Atmospheric Administration (NOAA). We calculate the seasonal mean AO and NAO index and find

that the correlation of ice volume flux against AO index (0.55) is higher than that against NAO index (0.34). Both of our study and Ricker et al. (2018) find that the AO may influence the sea ice export (2011-2016) more directly.

The CMST-based sea ice volume during both the melt season and the freezing season is first reported in this study. The estimations show that the mean ice volume export during the melt season is -750 (±120) km$^3$ which is about half of that during the freezing season (-1500±160 km$^3$). Annually, sea ice volume export (Figure 13) is -2250 km$^3$ and varies from -1970 (±290)

to -2490 (±280) km$^3$. It is verified again that the annual sea ice volume export through the Fram Strait does not show a significant growth or decline trend (Ricker et al., 2018; Spreen et al., 2009). And the maximum yearly ice volume export occurs in the year of 2012 while the ice volume export reaches its minimum in 2013. This decline in ice volume export derives from the decreases of both thickness and drift though the Fram Strait.

## 4 Discussions

The ensemble standard deviation (SD) map of CMST ice concentration, thickness and drift shows that uncertainties are larger downstream the east of Greenland (Figure 14). Therefore, following Krumpen et al. (2016) and Ricker et al. (2018), a different gateway over the Fram Strait that consists of a zonal gate and a meridional gate located at a slightly higher latitude comparing to previous studies is chosen (Kwok et al., 2004; Kwok and Rothrock, 1999; Spreen et al., 2009). Alternatively, the choice of lower latitude gate at 79°N (e.g., the ULS moored sites) is suggested to utilize the ULS thickness for rough volume flux

calculation. It should be noted that studies at different Fram Strait gates and over different periods will introduce deviations on the final ice volume estimation (Krumpen et al., 2016; Kwok et al., 2004; Kwok and Rothrock, 1999; Ricker et al., 2018; Spreen et al., 2009). For example, Ricker et al. (2018) investigated the sea ice flux in the Fram Strait and pointed out that the maximum (-540 km$^3$) occurs in March of 2011 and the minimum (-21 km$^3$) appears in February of 2011 from 2010 to 2017. However, on the different gate and period, Spreen et al. (2009) showed a relatively low maximum volume export of -420 km$^3$

and relatively high minimum flux (-92 km$^3$) in the freezing season.

We investigate the similar period with Ricker et al. (2018), but further extend the sea ice volume flux estimation to the melt season. Also, the CMST sea ice thickness data used in this study are evaluated to be reasonable when compared with in-situ observations (Mu et al., 2018a). The other important driver (sea ice drift) of ice volume export has also been compared with OSI SAF drift used in former estimations (Ricker et al., 2018) and Sentinel-1 SAR sea ice drift. The monthly mean CMST ice

drift show a better performance than NSIDC drift (Figure 1), and meanwhile, a good consistency is found between CMST and

OSI SAF (Figure 2d and Figure 7) drift. Overall, the estimation of volume export in this study reveals a reasonable sea ice volume export all year round.

The nearly 6 years' ice volume export through the Fram Strait is calculated and shown in Table 2. Besides the ice volume export (R) of Ricker et al. (2018), we also calculate the export using OSI SAF drift and CMST thickness (M1), and also using CMST thickness and drift (M2), respectively. It can be concluded that R is larger than M1 and M2 because R is derived from thicker CS2 thickness (Figure 5) and relatively faster OSI SAF drift (Figure 7). In addition, M1 is generally larger than M2 also due to the faster OSI SAF ice motion for most periods. There are also cases that M2 is larger than M1 when CMST has higher ice motion than OSI SAF, for example, in March, April and November of 2014. Ricker et al. (2018) gave their multi-year averaged volume export of -1711 $km^3$ in the freezing season. Our average estimate (M2) based on the CMST ice thickness and drift is -1580 $km^3$ while the volume flux (M1) derived from CMST thickness and OSI SAF drift is -1600 $km^3$ in the freezing season. The similar results between M1 and M2 are because that the CMST drift deviates minorly to OSI SAF drift in the cold seasons. But more reliable validations of CMST ice drift need more in-situ records and more systematic evaluations. To further validate the sea ice volume export in the melt season, we compare our CMST-based volume flux (M2) with the relative short-term summer ice volume flux that Krumpen et al. (2016) derived from airborne ice thickness and NSIDC ice drift data on the same export gates. The intercomparison shows that the sea ice volume export in August 2011 and July 2012 estimated by Krumpen et al. (2016) are smaller than this study. The underestimation of summer sea ice volume may deduce from a general underestimation of NSIDC drift during the melt season (Krumpen et al., 2016; Sumata et al., 2015; Sumata et al., 2014).

Through the Fram Strait gate located at ~79°N, Kwok et al. (1999 and 2004) investigated the summer sea ice export by using ULS thickness and area flux in the freezing season. The average annual ice volume flux is -2218 $km^3$ $yr^{-1}$ from 1991 to 1998 while the mean sea ice volume export from 1990 to 1995 is -2366 $km^3$ $yr^{-1}$ (Kwok et al., 2004; Kwok and Rothrock, 1999). To compare with previous studies (Kwok et al., 2004; Kwok and Rothrock, 1999; Vinje et al., 1998), we also calculate the sea ice volume flux through the same gate located at 79°N. Results (Figure 15) show that our annual mean sea ice volume export (-1352 $km^{-3}$) is smaller than these studies, which is expected because of the decline of sea ice thickness in recent decades. All these works show consistent seasonality with maximum export in March and minimum export in August. In a recent study, Wei et al. (2019) calculated the annual mean sea ice volume export (-3216 $km^3$ $yr^{-1}$) through the Fram Strait from their simulation during 1979 to 2012. Their estimations give a long period of sea ice volume export through the Fram Strait which can serve as an important reference when focusing on the long-term trend and the variations of the volume flux. However, this estimation seems to overestimate the volume flux owing to the overestimations of sea ice drift and thickness (Wei et al., 2019).

**5 Conclusions**

The CMST data over all seasons are first used to estimate ice volume export through the Fram Strait. Also, benefitting from the advantage of CMST data, the melt season (e.g. summer season and autumn season) ice volume export can be derived to

fill the satellite data gap over such periods. The entire seasonal and interannual variations of Arctic sea ice volume are helpful for communities that focus on climate teleconnection between Polar regions and low latitudes, Arctic freshwater transport and

ocean circulation. Conclusions of this study can be drawn as follows:

(1) The Arctic sea ice thickness and volume show a significant seasonal variation. The thickness and volume maximum usually occur in spring and the Arctic sea ice volume hits its minimum in autumn 2011 during the study period.

(2) Along the entire Fram Strait gate, the relative standard deviation (RSD) of ice drift (0.50) is higher than the RSD of ice thickness (0.30) and concentration (0.14), demonstrating that ice drift is a main driver of ice volume export through the Fram

Strait. The correlations of determination ($R^2$) also show that sea ice drift is a much more important contributor for sea ice volume export on monthly scale.

(3) The mean sea ice volume export during the melt season is around -750 ($\pm$120) $km^3$ which is about 50% of that during the freezing season (-1500$\pm$160 $km^3$). The lowest and largest annual sea ice volume export occur in 2013 and 2012, respectively. Seasonal sea ice volume export varies from -240 ($\pm$40) to -970 ($\pm$60) $km^3$, while the monthly sea ice export varies between -

11 $km^3$ (August of 2015) and -442 $km^3$ (March of 2011) during this study period. The abnormal ice volume export increase in autumn 2013 is primarily associated with the faster ice motion.

(4) The relative frequency (RF) of seasonal variation of CMST sea ice thickness shows that sea ice thicker than 2 m in spring and summer is more than that in other seasons. The summer mean ice drift that is lower than 6 km $d^{-1}$ is in the majority in each year.

An updated and improved CMST V2 sea ice data will be developed in the near future to obtain a long-term record for climate research.

*Data availability*. The CMST sea ice thickness and drift data are available at https://doi.pangaea.de/10.1594/PANGAEA.891475 and https://doi.pangaea.de/10.1594/PANGAEA.906973, respectively (Mu et al., 2018, last access: 1 October 2019). The OSI SAF drift data can be download at http://www.osi-saf.org/?q=content/sea-ice-products (last access: 1 January 2019). The latest released Polar Pathfinder Daily

25 km EASE-Grid sea ice drift data are provided by the National Snow and Ice Data Center (NSIDC, https://nsidc.org/data/nsidc-0116/versions/4, last access: 2 May 2019). The Sentinel-1 SAR sea ice drift data can be download at (https://www.ncbi.nlm.nih.gov/pmc/articles/PMC5999601/, last access: 7 Sep 2019). The helicopter-borne electromagnetic induction sounding (HEM) sea ice thickness are available at https://data.npolar.no/dataset/1ed8c57e-8041-42fd-95bb-cfe4e181e9b8 (last access: 3 May 2019). The AO and NAO index can be download at https://www.cpc.ncep.noaa.gov/products/precip/CWlink/daily_ao_index/ao.shtml

and https://www.cpc.ncep.noaa.gov/products/precip/CWlink/daily_ao_index/ao.shtml (last access: 8 September, 2019).

*Author contributions*. LM and QY conceptualized this study and provided the CMST sea ice data. CM conducted this study, performed the calculation and drafted the manuscript. RR supplied the sea ice data of Ricker et al. (2018) for intercomparison. LM, QY and RR polished this manuscript and improved the readability. QS, RW, BH and JL reviewed this manuscript.

*Competing interests.* The Authors declare that they have no conflict of interests.

*Acknowledgement.* This is a contribution to the Year of Polar Prediction (YOPP), a flagship activity of the Polar Prediction Project (PPP), initiated by the World Weather Research Programme (WWRP) of the World Meteorological Organization (WMO). Thanks are given to Yongwu Xiu, Ran Yang from School of Atmospheric Sciences, Sun Yat-sen University for the discussions. We also thank Yu Liang at Key laboratory of Marine Geology and Environment, Institute of Oceanology, Chinese Academy of Sciences for her advice and E. Hansen at Norwegian Polar Institute for providing the ULS data. This study is supported by the National Natural Science Foundation of China
(41922044, 41706224), National Key R&D Program of China (2018YFA0605901), the Opening fund of State Key Laboratory of Cryospheric Science (SKLCS-OP-2019-09), the Federal Ministry of Education and Research of Germany in the framework of SSIP (grant01LN1701A).

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

**Table 1.** OSI SAF drift data used in this study for comparison.

| Name | Product | Original data | Algorithm | Temporal resolution | Spatial Resolution | Period |
|---|---|---|---|---|---|---|
| OSI SAF | OSI-405 (merged) | SSMIS (91 GHz, DMSP F17), ASCAT (Metop-B), AMSR-2 (18.7 and 37 GHz) | CMCC | 2 days | 62.5 km | 2010-2016 |


**Table 2.** Monthly Arctic sea ice volume export through the Fram Strait in km$^3$ month$^{-1}$. Note that R is calculated by CS2 thickness and OSI SAF drift; M1 is calculated by CMST thickness and OSI SAF drift; and M2 is calculated by CMST thickness and CMST drift.

| | | Jan | Feb | Mar | Apr | May | Jun | Jul | Aug | Sep | Oct | Nov | Dec |
|---|---|---|---|---|---|---|---|---|---|---|---|---|---|
| 2010 | R | — | — | — | — | — | — | — | — | — | — | -227 | -275 |
| | M1 | — | — | — | — | — | — | — | — | — | — | -209 | -258 |
| | M2 | — | — | — | — | — | — | — | — | -148 | -222 | -195 | -239 |
| 2011 | R | -267 | **-21** | **-540** | -279 | — | — | — | — | — | -164 | -214 | -354 |
| | M1 | -238 | **-24** | **-478** | -255 | — | — | — | — | — | -149 | -163 | -293 |
| | M2 | -238 | **-34** | **-442** | -230 | -278 | -185 | -115 | -64 | -28 | -151 | -175 | -290 |
| 2012 | R | -129 | -381 | -379 | -487 | — | — | — | — | — | -203 | -182 | -187 |
| | M1 | -109 | -299 | -287 | -428 | — | — | — | — | — | -207 | -157 | -125 |
| | M2 | -137 | -300 | -267 | -372 | -334 | -218 | -187 | -131 | -100 | -160 | -149 | -136 |
| 2013 | R | -103 | -163 | -299 | -318 | — | — | — | — | — | -215 | -400 | -231 |
| | M1 | -80 | -122 | -254 | -254 | — | — | — | — | — | -212 | -372 | -211 |
| | M2 | -78 | -109 | -217 | -219 | -194 | -140 | -107 | -98 | -26 | -228 | -367 | -191 |
| 2014 | R | -78 | -195 | -345 | -452 | — | — | — | — | — | -200 | -165 | -373 |
| | M1 | -49 | -105 | -240 | -401 | — | — | — | — | — | -203 | -122 | -307 |
| | M2 | -61 | -114 | -282 | -425 | -232 | -161 | -112 | -184 | -194 | -170 | -162 | -283 |
| 2015 | R | -160 | -425 | -429 | -354 | — | — | — | — | — | -52 | -261 | -275 |
| | M1 | -129 | -358 | -328 | -284 | — | — | — | — | — | -72 | -215 | -243 |
| | M2 | -129 | -355 | -339 | -308 | -171 | -240 | -114 | **-11** | -107 | -78 | -192 | -244 |
| 2016 | R | -177 | -352 | -348 | -310 | — | — | — | — | — | -129 | -151 | -307 |
| | M1 | -129 | -272 | -255 | -264 | — | — | — | — | — | -98 | -90 | -243 |
| | M2 | -150 | -267 | -287 | -289 | -196 | -194 | -113 | -198 | -75 | -97 | -72 | -222 |


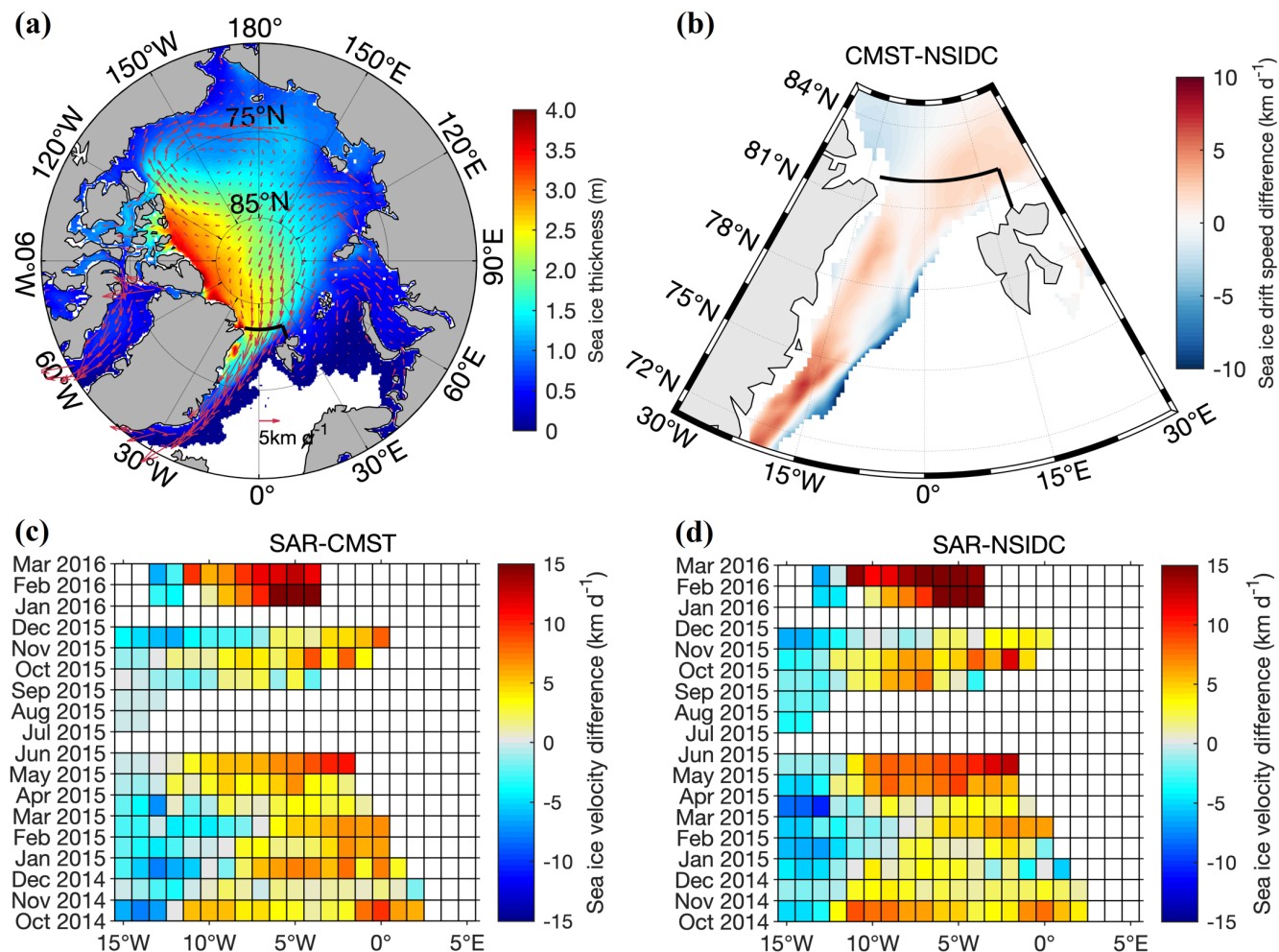

**Figure 1.** (a) The mean CMST sea ice drift and thickness averaged from September, 2010 to December, 2016. (b) The differences between CMST drift speed and NSIDC drift speed, the background color represents the magnitudes of ice velocity difference during the same period. The thick black line represents zonal and meridional sea ice export gates to derive sea ice

volume flux through the Fram Strait. (c) Meridional velocity difference between SAR drift and CMST drift at the Fram Strait (79 °N). (d) Meridional velocity difference between Sentinel-1 SAR drift and NSIDC drift at the Fram Strait (79 °N).

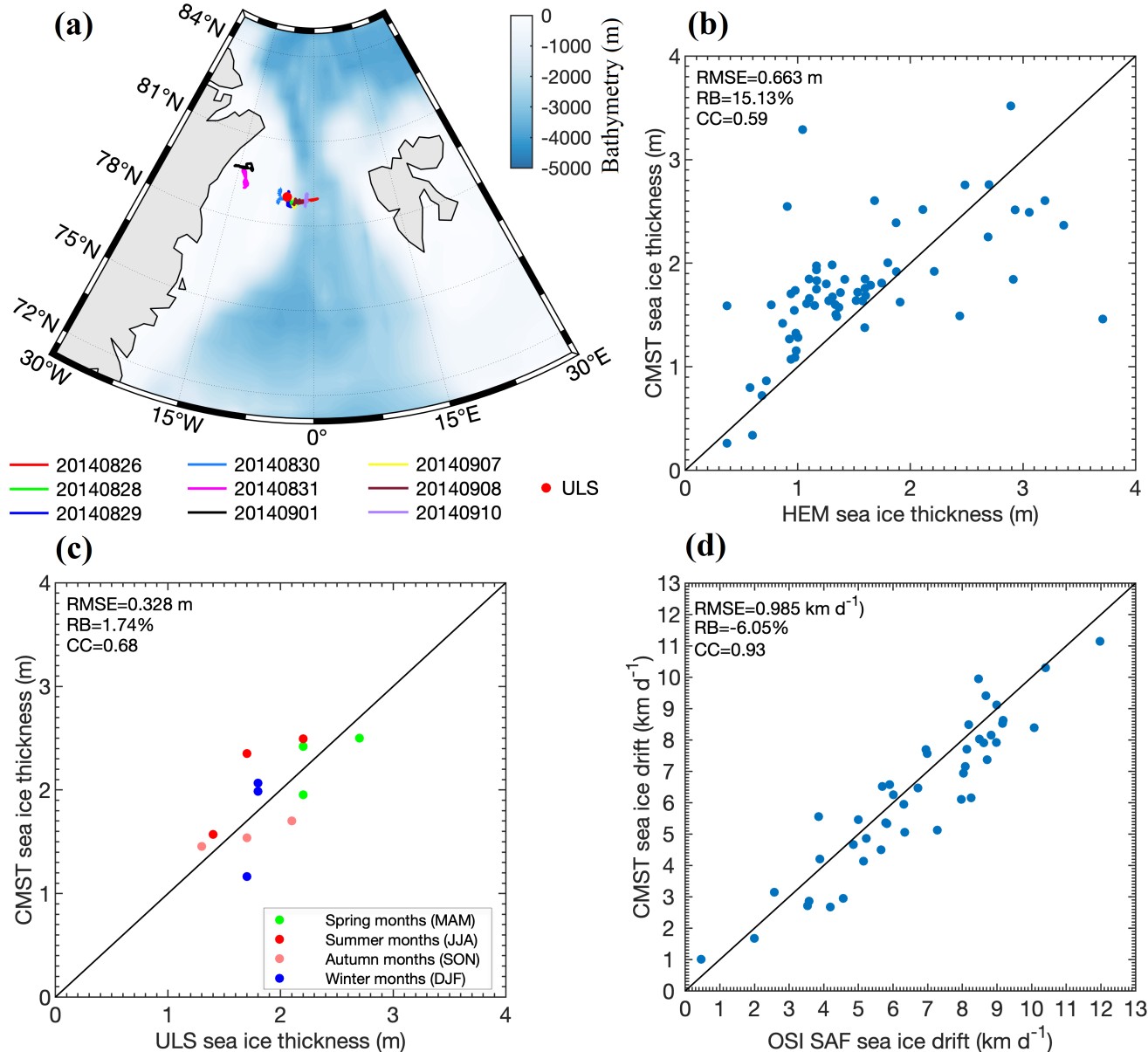

**Figure 2.** (a) The trajectories of 9 separate flights of HEM measurement campaigns carried out in the Fram Strait and the red dot denotes the site of ULS, Scatter plots of (b) daily mean sea ice thickness derived from CMST and HEM data, (c) monthly mean sea ice thickness derived from CMST and ULS data, (d) monthly average sea ice drift based on CMST and OSI SAF.

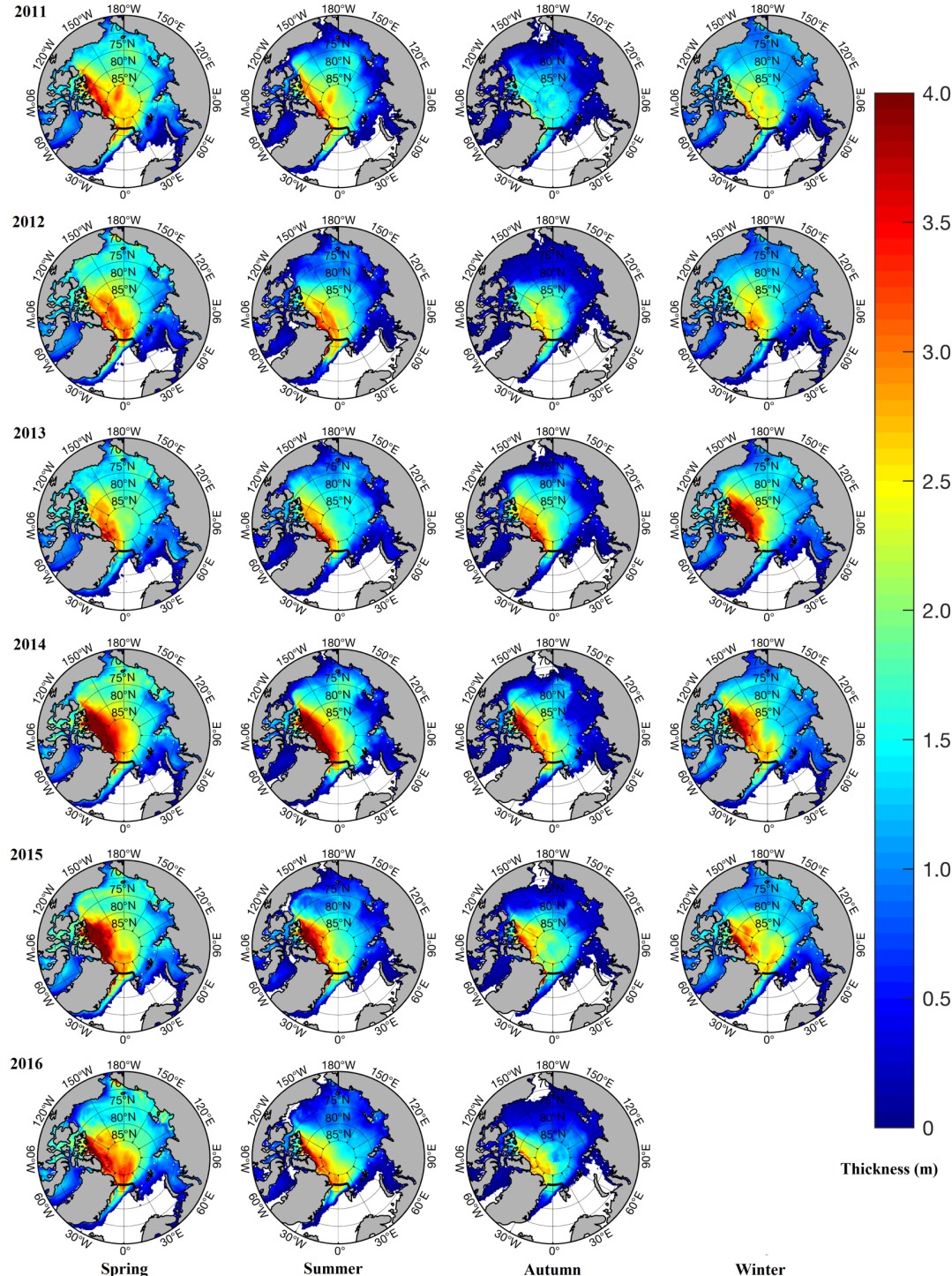

**Figure 3.** Seasonal variation of Arctic sea ice thickness. The thick black line represents the sea ice fluxgate in the Fram Strait used in this study.

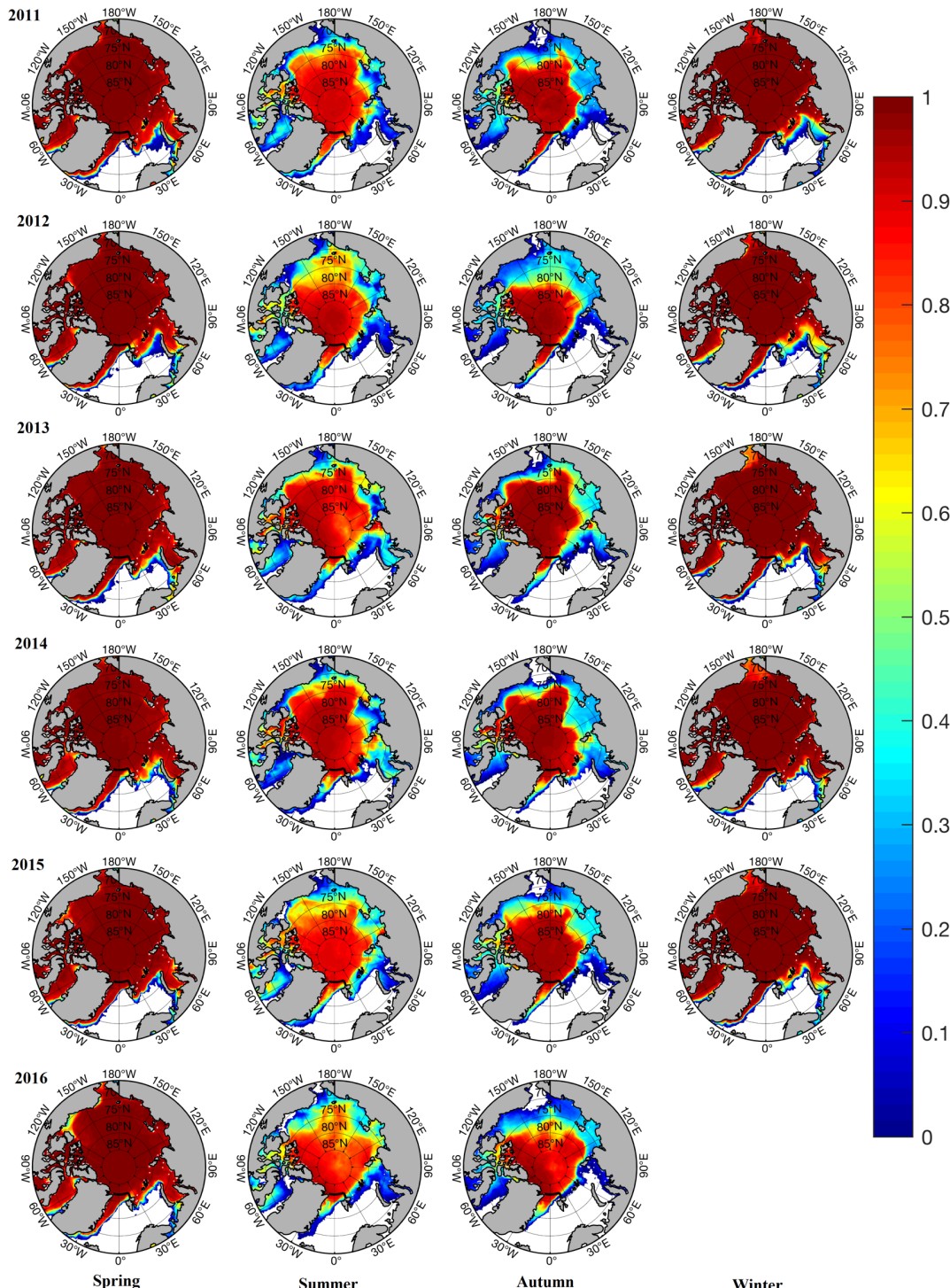

**Figure 4.** Seasonal variation of Arctic sea ice concentration. The thick black line represents the sea ice fluxgate in the Fram Strait.

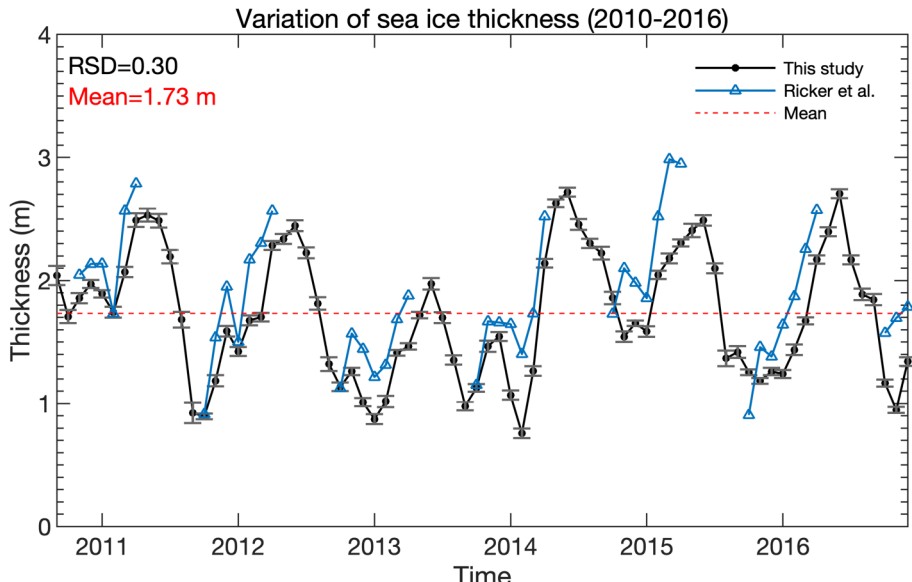

**Figure 5.** CMST sea ice thickness averaged over the entire Fram Strait gate, from September 2010 to December 2016. The black dotted line denotes monthly mean ice thickness based on CMST data with corresponding standard deviations while the blue dotted line represents monthly mean effective sea ice thickness of Ricker at al. (2018).

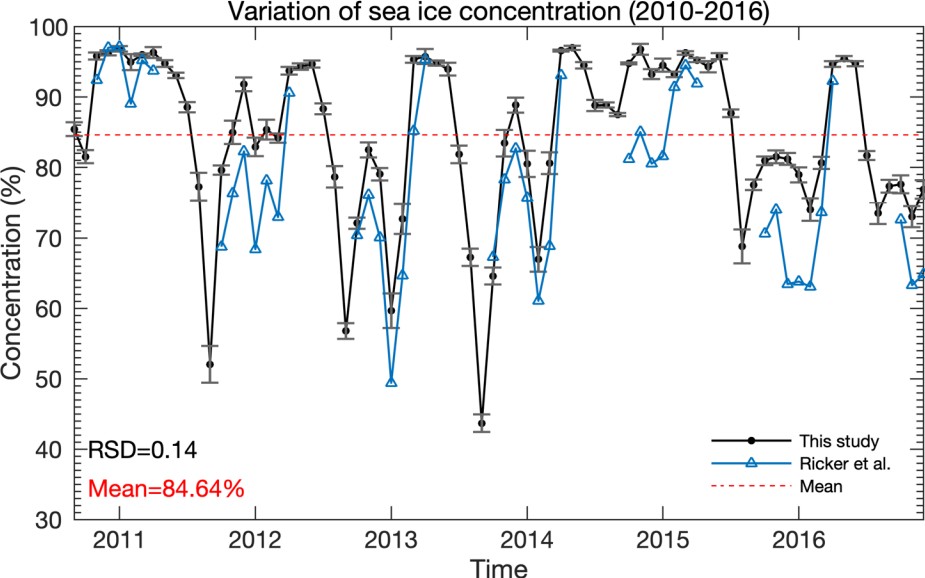

**Figure 6.** CMST sea ice concentration averaged over the entire Fram Strait gate, from September 2010 to December 2016. The black dotted line represents monthly mean ice concentration based on CMST data with corresponding standard deviations while the blue dotted line represents monthly mean ice concentration of OSI SAF.

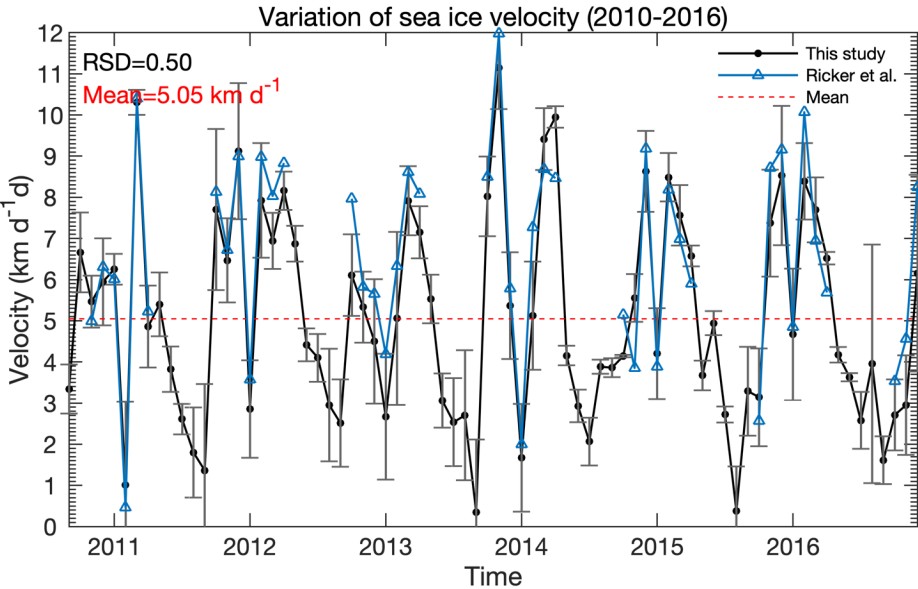


**Figure 7.** CMST sea ice drift averaged over the entire Fram Strait gate, from September 2010 to December 2016. The black dotted line represents monthly mean ice drift based on CMST data with corresponding standard deviations while the blue dotted line shows the monthly mean ice drift of OSI SAF.

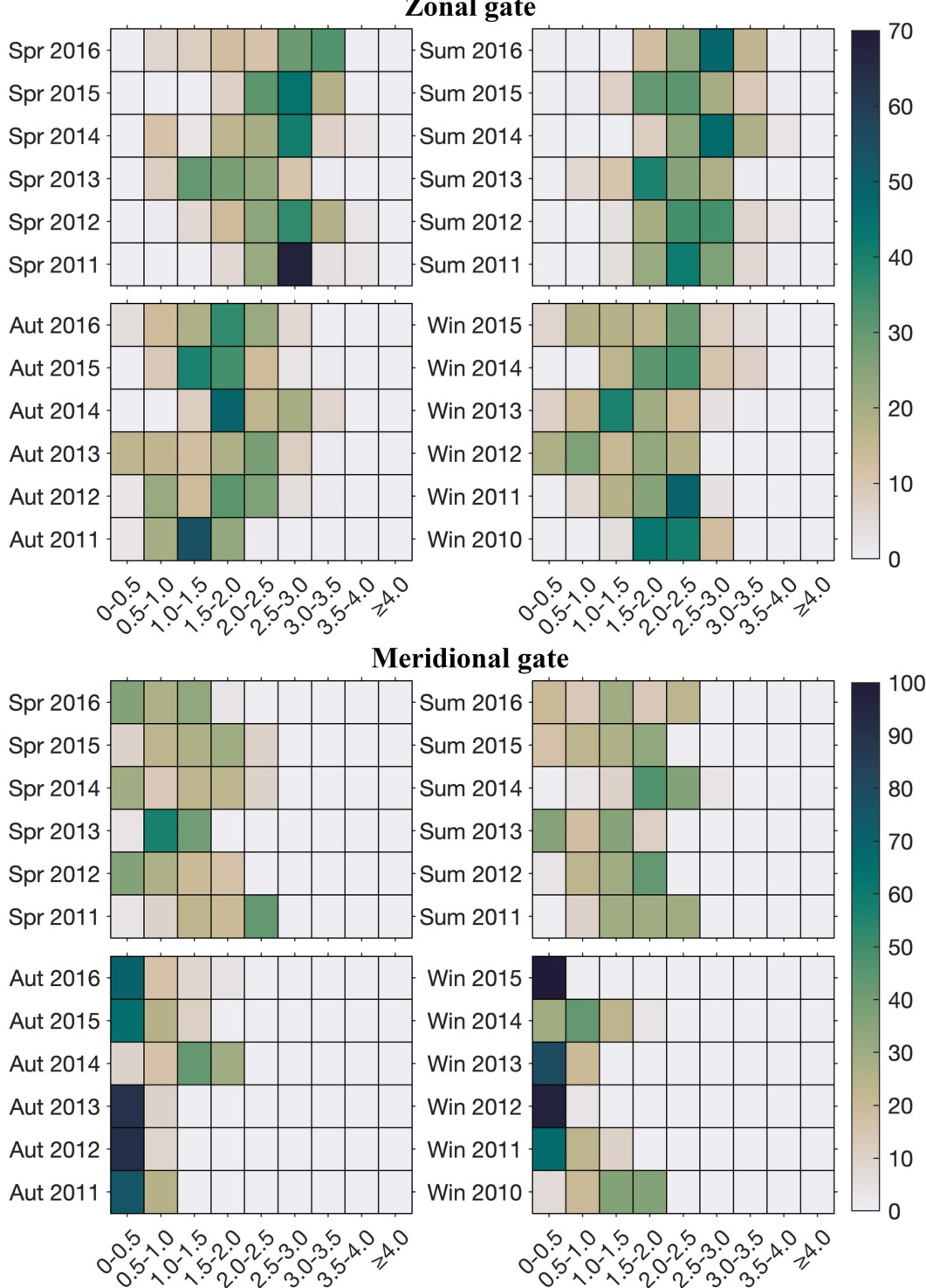

**Figure 8.** Seasonal variation of relative frequency (unit: %) of CMST sea ice thickness (unit: m) over the Fram Strait gate.


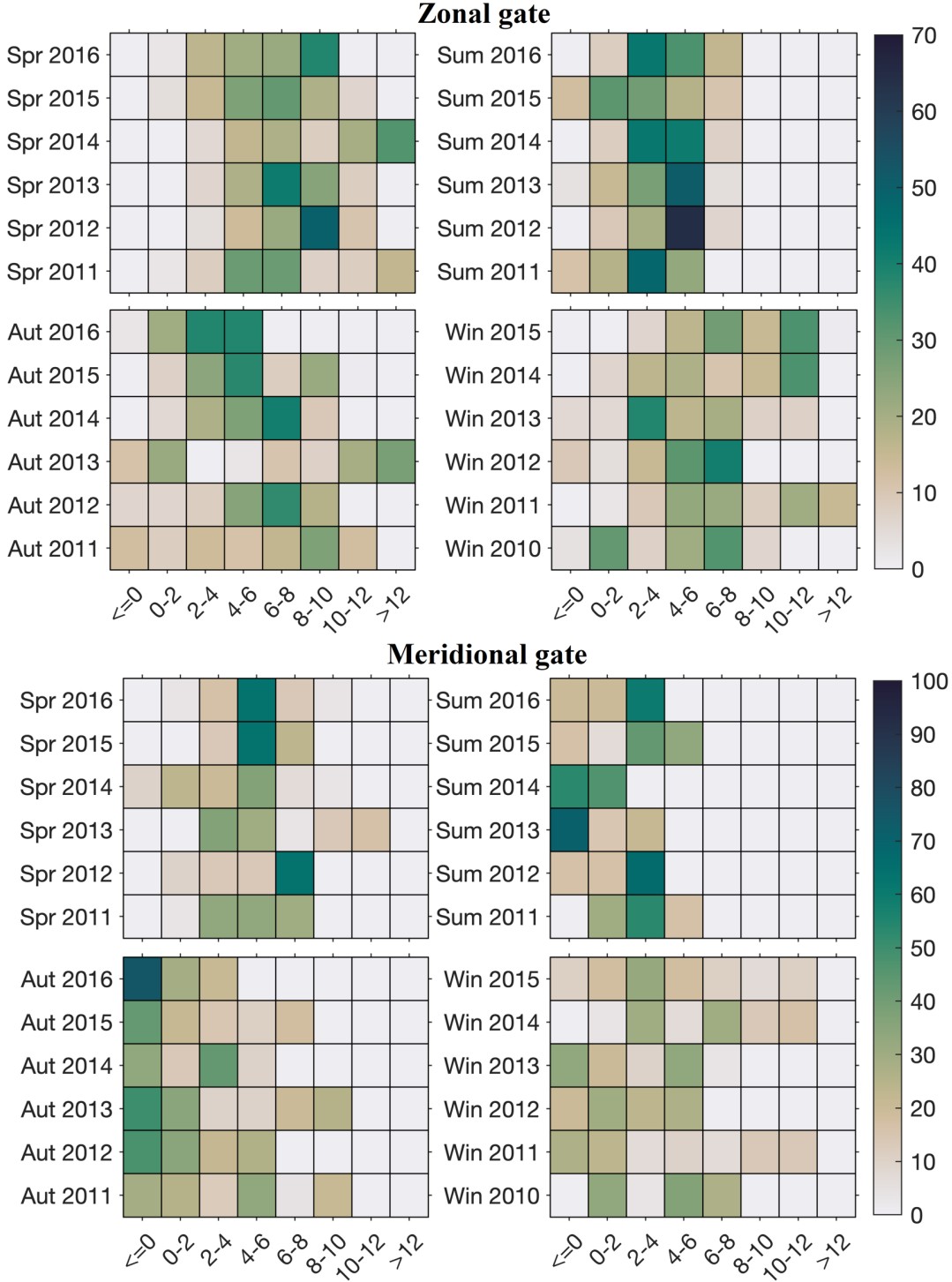

**Figure 9.** Seasonal variation of relative frequency (unit: %) of CMST sea ice drift (unit: km d$^{-1}$) over the entire Fram Strait gate.

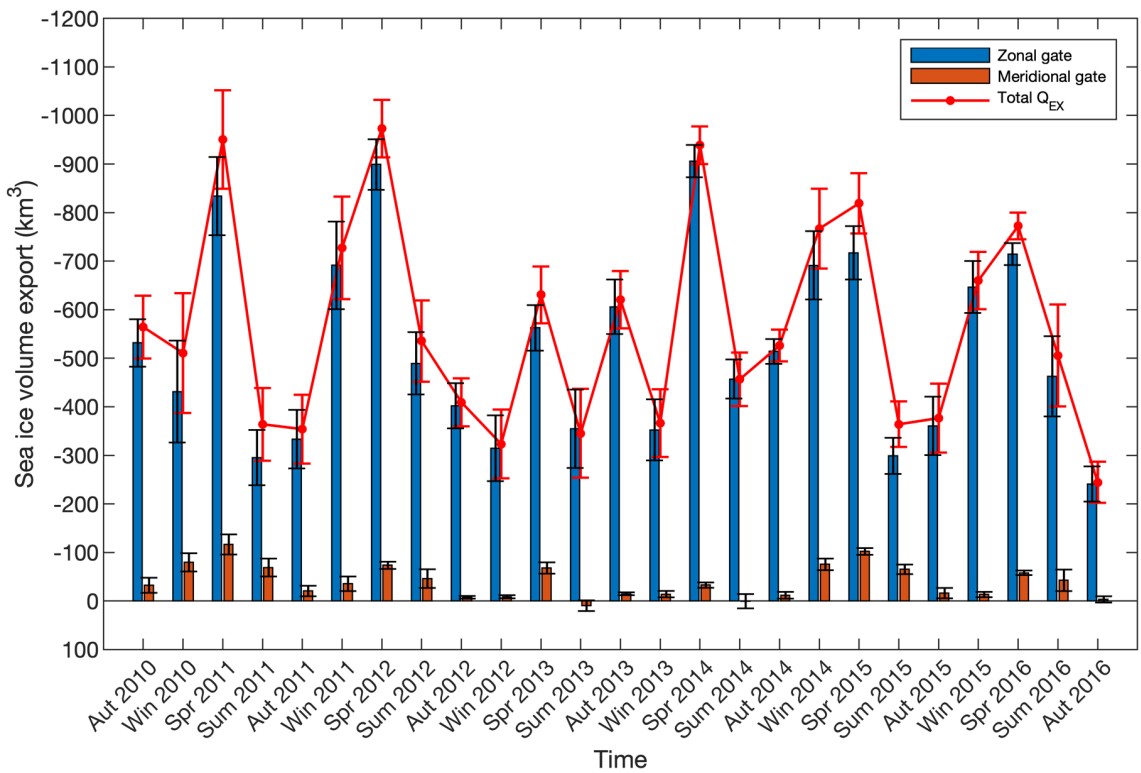


**Figure 10.** CMST seasonal Arctic sea ice volume export (unit: km$^3$) through the Fram Strait with corresponding uncertainty. Q$_{EX}$ represents the sea ice volume export based on CMST thickness and drift (similarly hereinafter).

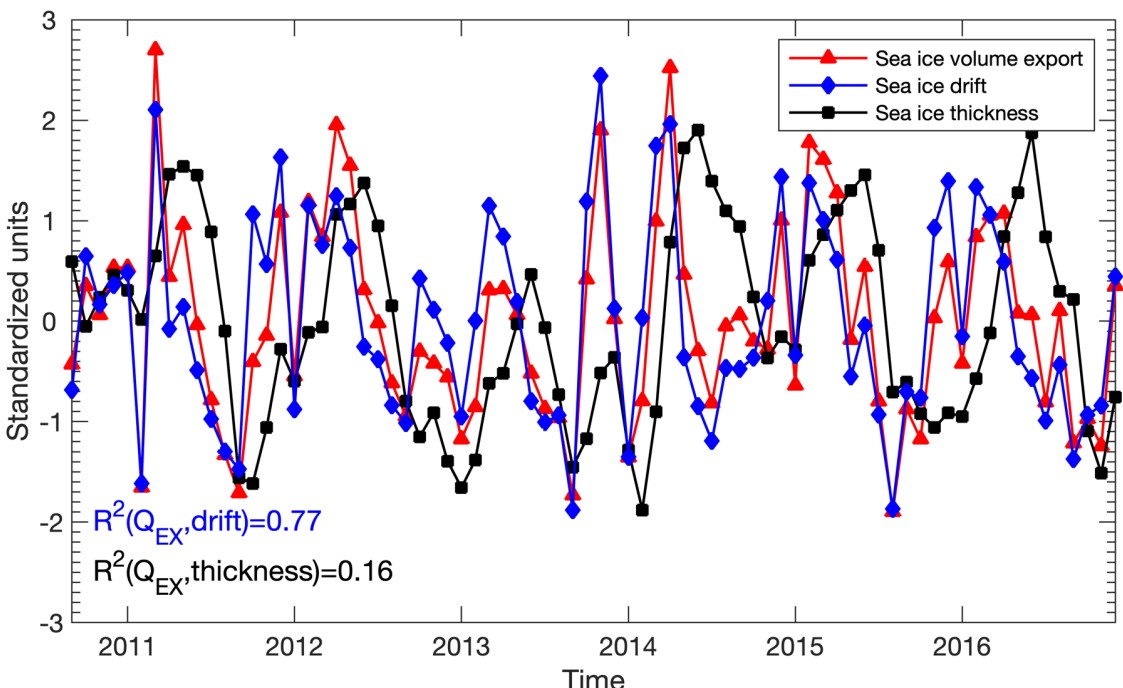

**Figure 11.** Time series of standardized monthly mean sea ice volume export ($Q_{EX}$, red line) and corresponding monthly mean sea ice drift (blue line) and sea ice thickness (black line), including correlation of determination ($R^2$).

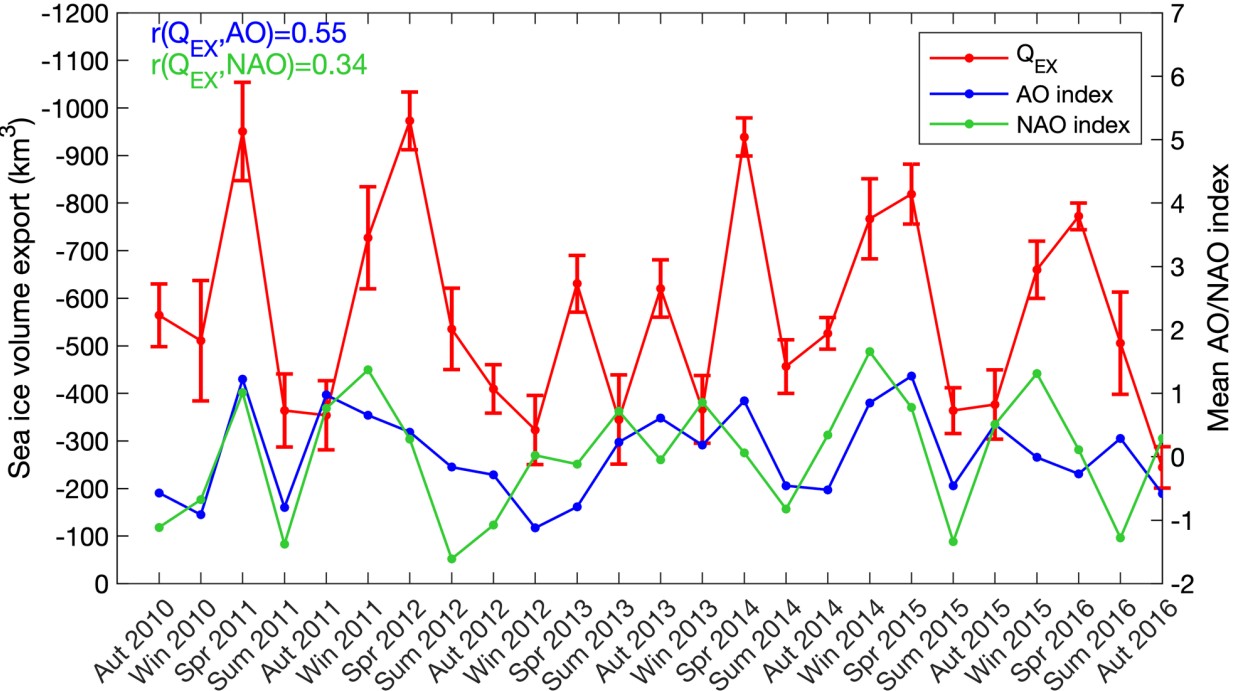

**Figure 12.** Time series of seasonal mean sea ice volume export (Q$_{EX}$, unit: km$^3$, red line) and corresponding mean seasonal AO (blue line) and NAO (green line) index, including correlation coefficient (r).

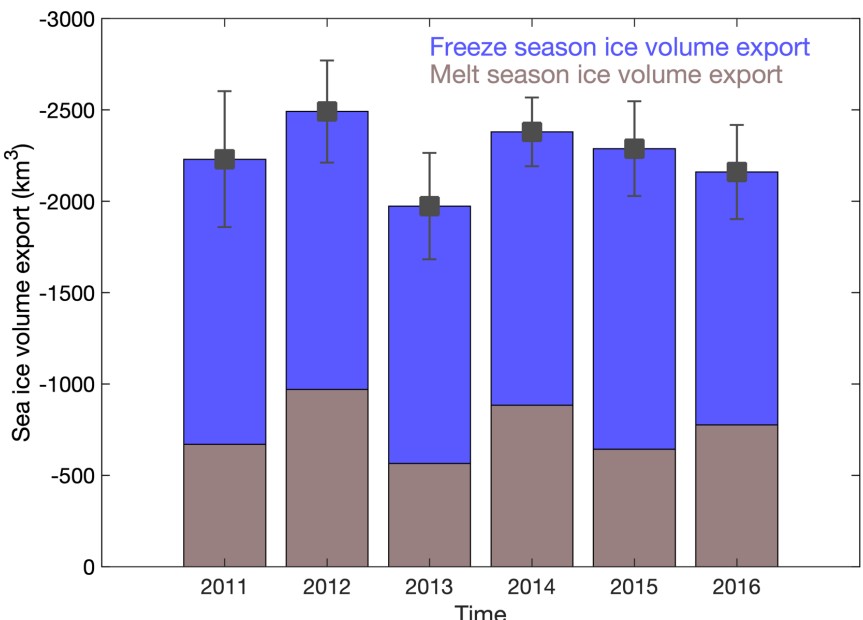

**Figure 13.** CMST interannual Arctic sea ice volume export (unit: km$^3$) through the entire Fram Strait with corresponding uncertainty. The freezing season represents the months from October to April and the melt season is during May to September.

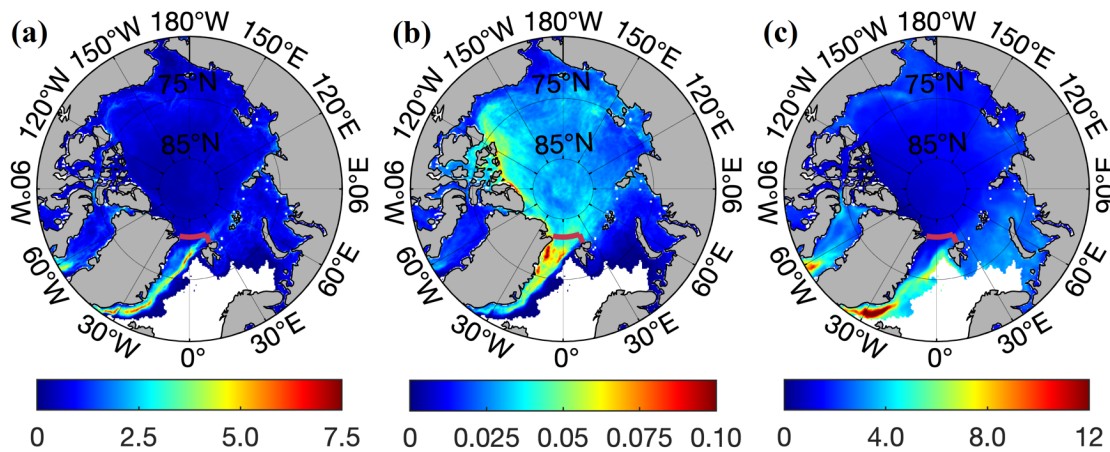

**Figure 14.** The mean ensemble standard deviation (SD) map of CMST (a) sea ice concentration (unit: %), (b) sea ice thickness (unit: m) and (c) sea ice drift (unit: km d$^{-1}$) from September, 2010 to December, 2016. The thick red line represents zonal and meridional sea ice export gates to derive sea ice volume flux through the Fram Strait.

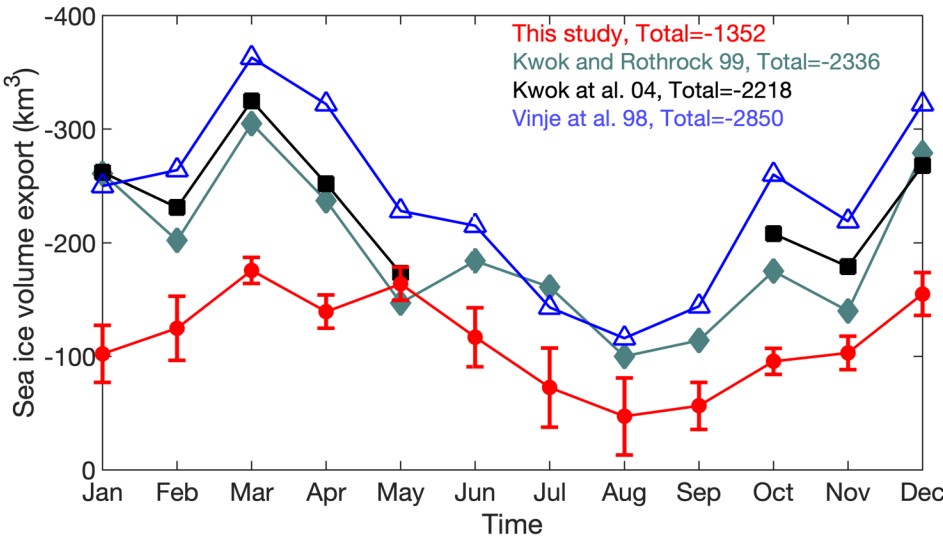


**Figure 15**. Monthly mean sea ice volume export (unit: km³) at 79°N transect in the Fram Strait from this study (red line), Kwok and Rothrock (1999, dark green line), Kwok et al. (2004, black line) and Vinje et al. (1998, blue line).