# Peer review of "Sea ice export through the Fram Strait derived from a combined model and satellite data set"

_The Cryosphere, 2019_

## Referee Comment (RC1) · Anonymous Referee #1 · 21 Aug 2019

Overall:

In this manuscript the authors use a recently published CMST data product on Arctic sea ice thickness and drift to derive and analyze a variability of Fram Strait sea ice export over the period of 2010-2016.

The manuscript is clearly written and results are also presented well. The provided estimates of melt season volume fluxes is definitely a strong side of the entire story.

My only moderate (to major) concern is related with the use of the 82N gate alone to calculate the sea ice export from the Arctic. Though a resulted lower uncertainty as a main motivation (presented in Fig. 12) is apparent, given their new data product, it should be relatively straightforward and highly beneficial for this work to run the same

computations for the 79N transect. This will allow to consider these new results in the context of earlier/similar studies on the topic, e.g. Kwok et al., 2004, Spreen et al., 2009. The authors already refer to Kwok et al., 2004 when discussing and comparing the seasonal/annual volume fluxes and their changes over two decades; a more direct comparison of the two products with respect to the volume fluxes would be possible if the fluxes at 79N were presented as well. Therefore, in addition to their flux estimates for the northern gates, the authors are encouraged to extend their calculations to a southerly transect at 79N which accommodates the FS ULSs.

In general, the manuscript deserves to be published after these relatively moderate modifications to the content. Some minor corrections are also suggested in the list of minor comments below.

Minor comments:

Line 20: please round off to significant figures throughout the text, e.g. 244+-43 can well be rounded to 240+-40 etc

Line 51: "In terms of sea ice volume flux, Ricker et al. (2018) and Bi et al. (2018)..." one can site Zamani et al., 2019 (https://doi.org/10.1007/s00382-019-04699-z) too.

Lines 75 – 88. Pease indicate in 2.1. that CMST data in addition to ice thickness and concentration also comprises the modelled/assimilated ice drift velocities.

Line 106: Please cite HEM data properly, refer to Renner at al., 2014 (10.1002/2014GL060369) and a respective data citation found in (https://data.npolar.no/dataset/1ed8c57e-8041-42fd-95bb-cfe4e181e9b8)

Line 133: please state explicitly "meridional velocity" and "zonal velocity" or "meridional and zonal components of the sea ice drift vector"

Line 181: "thinning trend from west to east", one can mention this is in line with other studies on the topic (e.g. Kwok et al., Hansen et al., Krumpen et al., etc).

Line 187: "within the Arctic basin"?

Line 197: "...slightly smaller than that of CS2". This is certainly not the case; the discrepancies between the data sets as presented in Figure 5 can not be referred to as "slightly smaller".

Line 209: "...consistent with previous studies, such as..." pointed already in Kwok, et al.

Sections 3.3 and 3.4. As I pointed earlier, having sea ice volume fluxes estimated at 79N too, would have made a comparison with other studies much more straightforward. No need to add more figures, the additional data can well be accommodated in the existing ones.

---

## Referee Comment (RC2) · Anonymous Referee #2 · 23 Aug 2019

General comments:

This paper extended the estimations of sea ice export through the Fram Strait to the melting seasons using the combined model and satellite thickness (CMST) data set, which assimilates CryoSat-2 and SMOS thickness products, as well as sea ice concentration data. The paper is well written and easy to follow. The CMST data and methods to calculate sea ice export are already published in previous studies. The novelty of this paper might be the sea ice export during the melt season. I expect more elaborate analysis and discussion about the uncertainties of the input data during the melt season, and their impact on the estimation of sea ice export. I suggest to consider this manuscript for further publication after more analysis and comparisons are presented in the revised version.

[Figure]

Specific comments:

P5 L126-128: What kind of interpolation method is used here? Section 3.1 validation of CMST data: Validation of sea ice drift: Since sea ice drift is the essential parameter in the calculation of sea ice flux, more comparison and analysis should be carried out about the uncertainty of CMST sea ice drift product, especially during the melt season where the uncertainty of sea ice drift is greatest. From fig. 1b, it is difficult to figure out how well CMST sea ice drift data fits to the NSIDC data along the gates. Why do you use NSIDC here for the comparison, not OSI SAF? P6L160-161 please add a short description about the domain (is it the Fram Strait gate?), time period used, etc. P6L155: what do you mean with "within the current understanding"? P7 L204: "moderate decline" during the 6 years? P7 L212-213: Please define the relative frequency used here. Fig. 8 and 9: It might be helpful to add the mean ice thickness/ice drift of each season in the figures. P7 L214-218: Could you explain why ice thickness is thickest in summer and ice drift is slowest in summer? P8 L224-225: better to mention this earlier in section 2.6. Section 3.3 Sea ice volume export through the Fram Strait: It might be interesting to analyze the respective contributions of ice drift and ice thickness to the seasonal variation of sea ice export. How many percent of sea ice export variation can be explained by ice drift/ice thickness? More discussions could be made about the causes of the variations of ice drift, ice thickness and ice export during the melt season, e.g. what is the reason of the extreme low ice export in August 2015? Authors could also do some additional comparisons with atmospheric circulation variability, e.g. the Arctic Oscillation indices. P8 L243: 1503±158km3 minus sign is missing. P10 L286: the values should have minus signs? P11 L315-316 and L319-320 are the same sentences

---

## Author Comment (AC1) · 23 Sep 2019

**General Comments:**

In this manuscript the authors use a recently published CMST data product on Arctic sea ice thickness and drift to derive and analyze a variability of Fram Strait sea ice export over the period of 2010-2016.

The manuscript is clearly written and results are also presented well. The provided estimates of melt season volume fluxes is definitely a strong side of the entire story.

My only moderate (to major) concern is related with the use of the 82°N gate alone to calculate the sea ice export from the Arctic. Though a resulted lower uncertainty as a main motivation (presented in Fig. 12) is apparent, given their new data product, it should be relatively straightforward and highly beneficial for this work to run the same computations for the 79°N transect. This will allow to consider these new results in the context of earlier/similar studies on the topic, e.g. Kwok et al., 2004, Spreen et al., 2009. The authors already refer to Kwok et al., 2004 when discussing and comparing the seasonal/annual volume fluxes and their changes over two decades; a more direct comparison of the two products with respect to the volume fluxes would be possible if the fluxes at 79°N were presented as well. Therefore, in addition to their flux estimates for the northern gates, the authors are encouraged to extend their calculations to a southerly transect at 79°N which accommodates the FS ULSs.

In general, the manuscript deserves to be published after these relatively moderate modifications to the content. Some minor corrections are also suggested in the list of minor comments below.

Dear Reviewer:

We would like to thank you for the helpful comments to improve this manuscript. We agree that providing the analysis for the 79°N gate is beneficial for the community since it has been widely used in former studies. Therefore, we added calculations for the 79°N gate and further provide the volume flux to address the reviewer's concern. Then, we further compared our sea ice drift data from CMST with that derived from Sentinel-1 SAR images, which confirmed that CMST data have smaller errors than NSIDC sea ice drift data in the melt season. And we replotted the previous figures of the relative frequency using heat map for better visibility in our manuscript with tracks.

The specific response and revisions are shown below. They are in blue font for clarity.

**Specific comments:**

**Point 1:** Line 20: please round off to significant figures throughout the text, e.g. 244+-43 can well be rounded to 240+-40 etc.

*Response 1:* The statistics are rounded off to significant figures throughout the text in interannual and seasonal time scales. To get a better review on sea ice volume export between different months and different products (*e.g., table 2 in our manuscript with tracks*), we preserved the previous figures since the monthly sea ice volume export is relatively smaller compared to seasonal and interannual sea ice export. For example, the sea ice volume export in September 2016 (-75 km$^3$) is only -3 km$^3$ larger than that in November 2016 (-72 km$^3$). If both are rounded off in monthly time scale, their difference will reach -10 km$^3$ which is more than two times of -3 km$^3$. And the difference between R, M1 and M2 (table 2) could also been smoothed.

**Point 2:** Line 51: "In terms of sea ice volume flux, Ricker et al. (2018) and Bi et al. (2018)…" one can site Zamani et al., 2019 (https://doi.org/10.1007/s00382-019-04699-z) too.

*Respond 2:* We realized the constructive work from Zamani et al., 2019 and cited this paper in our manuscript. *(please see P2 line 54 in our manuscript with tracks)*

**Point 3:** Lines 75 – 88. Pease indicate in 2.1. that CMST data in addition to ice thickness and concentration also comprises the modelled/assimilated ice drift velocities.

*Respond 3:* We have indicated that the CMST data provide sea ice thickness, concentration and ice drift data as suggested. The CMST sea ice drift data have been submitted to PANGAEA, and are under processing with ID: PDI-21565. *(please see P3 line 78-79 in our manuscript with tracks)*

**Point 4:** Line 106: Please cite HEM data properly, refer to Renner at al., 2014 (10.1002/2014GL060369) and a respective data citation found in (https://data.npolar.no/dataset/1ed8c57e-8041-42fd-95bb-cfe4e181e9b8).

*Respond 4:* Corrected. And the website for publishing the data has been added in our manuscript. *(please see P4 line 121-122 and 128 in our manuscript with tracks)*

**Point 5:** Line 133: please state explicitly "meridional velocity" and "zonal velocity" or "meridional and zonal components of the sea ice drift vector".

*Respond 5:* As suggested above, we have stated explicitly that we use the meridional velocity along the zonal gate and zonal velocity along the meridional gate. *(please see P5 line 153 in our manuscript with tracks)*

**Point 6:** Line 181: "thinning trend from west to east", one can mention this is in line with other studies on the topic (e.g. Kwok et al., Hansen et al., Krumpen et al., etc).

*Respond 6:* Agreed. We added this sentence *in P8 line 219-220 in our manuscript with tracks*.

**Point 7:** Line 187: "within the Arctic basin"?

***Respond 7:*** Agree with this description. To avoid confusion, we used the description that referee suggested. *(please see P8 line 225 in our manuscript with tracks)*

**Point 8:** Line 197: "...slightly smaller than that of CS2". This is certainly not the case; the discrepancies between the data sets as presented in Figure 5 can not be referred to as "slightly smaller".

***Respond 8:*** We agreed on this comment, and we deleted the word 'slightly'. *(please see P8 line 235 in our manuscript with tracks)*

**Point 9:** Line 209: ". . .consistent with previous studies, such as. . ." pointed already in Kwok, et al.

***Respond 9:*** We refined this sentence to '*It is shown that the ice drift with maximal RSD is more likely to affect variations in sea ice volume flux, which is corresponding to the previous findings in Kwok et al. (1999), Ricker et al. (2018) and Bi et al. (2018).*'. *(please see P8 line 248-250 in our manuscript with tracks)*

**Point 10:** Sections 3.3 and 3.4. As I pointed earlier, having sea ice volume fluxes estimated at 79°N too, would have made a comparison with other studies much more straightforward. No need to add more figures, the additional data can well be accommodated in the existing ones.

***Respond 10:*** We realized this is really a constructive suggestion. It is worth to compare our sea ice volume flux with Vinje et al. (1998), Kwok et al. (1999), and Kwok et al. (2004) at 79°N. The monthly mean sea ice export along the outlet at 79°N is shown in below Figure 1 (*also shown in Figure 15 in our manuscript with tracks*). Result shows that our annual mean sea ice volume export is smaller than previous studies (Vinje et al. 1998, Kwok et al. 1999, Kwok et al. 2004), which is expected because of the decline of sea ice thickness in recent decade. All these works show consistent seasonality with maximum export in March and minimum export in August. The sea ice volume flux from Spreen et al. (2009) suggested by the reviewer is not shown because it was calculated at 80°N or 76°N. *(please see P12 line 365-374 in our manuscript with tracks)*

[Figure]

Figure 1. Mean monthly sea ice volume export (unit: $km^3$) at 79°N transect in the Fram Strait from this study (red line), Kwok et al. (1999, dark green line), Kwok et al. (2004, black line) and Vinje et al. (1998, blue line).

[revised manuscript text omitted]
 2011). The RF of sea ice thickness along the meridional gate also shows the major fractions appearing in the spring and summer. In addition, the values of seasonal mean sea ice thickness are 2.06 m for spring, 2.11m for summer, 1.32 m for autumn and 1.43 m for winter over the entire outlet, respectively.Figure 9approximately90),The seasonal mean sea ice velocity over the entire gate is larger than 5 km d$^{-1}$ except the 3 km d$^{-1}$ during summer.~~ And it can be found that the spring and winter ice concentration along the zonal gate is larger than that of summer and autumn.

**3.3 Sea ice volume export through the Fram Strait**

[revised manuscript text omitted]

---

## Author Comment (AC2) · 23 Sep 2019

**General Comments:**

This paper extended the estimations of sea ice export through the Fram Strait to the melting seasons using the combined model and satellite thickness (CMST) data set, which assimilates CryoSat-2 and SMOS thickness products, as well as sea ice concentration data. The paper is well written and easy to follow. The CMST data and methods to calculate sea ice export are already published in previous studies. The novelty of this paper might be the sea ice export during the melt season. I expect more elaborate analysis and discussion about the uncertainties of the input data during the melt season, and their impact on the estimation of sea ice export. I suggest to consider this manuscript for further publication after more analysis and comparisons are presented in the revised version.

Dear Reviewer:

   We would like to thank you for the helpful comments to improve this manuscript. We further compared the sea ice drift data from CMST with that derived from Sentinel-1 SAR images, which confirmed that CMST data have smaller errors than NSIDC sea ice drift data in the melt season. And we replotted the previous figures of the relative frequency using heat map for better visibility. To compare with other studies much more straightforward, we also added calculations for the 79°N gate and further provide the volume flux comparison in our manuscript with tracks.

   Below, we repeat each comment and insert our replies in the text where revisions were made. All responses are in blue font for clarity of reading.

**Specific comments:**
**Point 1:** P5 L126-128: What kind of interpolation method is used here?
*Response:* We use linear interpolation method to interpolate the sea ice data onto the outlet. And we added this point in our manuscript. *(please see P5 line 145-146 in our manuscript)*

**Point 2:** Section 3.1 validation of CMST data: Validation of sea ice drift: Since sea ice drift is the essential parameter in the calculation of sea ice flux, more comparison and analysis should be carried out about the uncertainty of CMST sea ice drift product, especially during the melt season where the uncertainty of sea ice drift is greatest. From fig. 1b, it is difficult to figure out how well CMST sea ice drift data fits to the NSIDC data along the gates. Why do you use NSIDC here for the comparison, not OSI SAF?

***Response:*** We agreed that more comparison and analysis about CMST drift data are necessitated. *(please P4 section 2.4 and P6 section 3.1 line 174-191)*

(1) The reason why we choose NSIDC drift for comparison is that during this study period (from September 2010 to December 2016) only the NSIDC sea ice drift data covers all of the melt seasons, while the OSISAF sea ice drift data are only available in the freezing season. The 6 years' mean difference between CMST sea ice drift data and NSIDC drift data over the years would be more informative. In addition, we have compared the sea ice drift over the entire Fram Strait gate from CMST with that from OSISAF used in Ricker et al. (2018) in the previous manuscript (also *shown in Figure 2d and Figure 7 in our manuscript with tracks*). The comparison (Figure 1a and 1b below, *also shown in Figure 2d and Figure 7 in our manuscript with tracks*) shows that the CMST drift fits the OSISAF drift pretty well.

(2) Owing to the purpose of our manuscript is estimating the sea ice volume export through the Fram Strait, we then further compared the CMST drift data with a high-resolution Sentinel-1 SAR drift data (Muckenhuber et al., 2016; Muckenhuber et al., 2018). The Sentinel-1 SAR drift data are calculated as monthly mean sea ice velocities at 79°N from October 2014 to February 2016, and cover both melt season and freezing season. To show the performance of CMST sea ice drift data in the Fram Strait, the NSIDC sea ice drift data are also compared with the SAR drift data. Results (Figure 2c and 2d below, *also shown in Figure 1c and 1d in our manuscript with tracks*) show that the CMST data exhibit even smaller errors than NSIDC data.

(3) For better showing the difference between CMST drift data and NSIDC data, we removed the Figure 1b in the previous version of our manuscript and showed the 6 years' mean difference (Figure 2b below, also shown in Figure 1b in our manuscript with tracks) specifically in the Fram Strait where this study focuses on.

[Figure]

Figure 1. (a) Scatter plots of monthly average sea ice drift based on CMST and OSI SAF. (b) CMST sea ice drift averaged over the entire Fram Strait gate, from September 2010 to December 2016. The black dotted line represents monthly mean ice drift based on CMST data with corresponding standard deviations while the blue dotted line shows the monthly mean ice drift of OSI SAF.

[Figure]

Figure 2. (a) The mean CMST sea ice drift and thickness averaged from September, 2010 to December, 2016. (b) The differences between CMST drift and NSIDC drift, the background color represents the magnitudes of ice velocity difference during the same period. The thick black line represents zonal and meridional sea ice gates to derive sea ice volume flux through the Fram Strait. (c) Meridional velocity difference between SAR drift and CMST drift at the Fram Strait (79 °N). (d) Meridional velocity difference between Sentinel-1 SAR drift and NSIDC drift at the Fram Strait (79 °N).

**Point 3:** P6L160-161 please add a short description about the domain (is it the Fram Strait gate?), time period used, etc.

*Response:* The monthly mean CMST drift data are calculated over the entire Fram Strait gate defined in this study. And it is compared with the OSI SAF sea ice drift data used in Ricker et al., (2018) within the same period from September 2010 to December 2016 and same domain defined before. We further added this information in the manuscript. *(please see P7 line 196-198 in our manuscript with tracks)*

**Point 4:** P6L155: what do you mean with "within the current understanding"?

*Response:* We rewrote our sentence as '*The mean sea ice thickness is distributed as expected (Tilling et al., 2015; Kwok et al., 2018), i.e., the relatively thicker ice, which is more than 2.5 m, mainly distributes in the north of Greenland and the Canadian Arctic Archipelago and the sea ice becomes thinner towards the Eurasia coasts. (Figure 1a)*'. *(please see P6 line 172-174 in our manuscript with tracks)*

**Point 5:** P7 L204: "moderate decline" during the 6 years?

*Response:* We realized that it is ambiguous previously. We changed the sentence to

*'The analysis of ice concentration shows a steadily low values in the melt season'.*
*(please see P8 line 242-243 in our manuscript with tracks)*

**Point 6:** P7 L212-213: Please define the relative frequency used here. Fig. 8 and 9: It might be helpful to add the mean ice thickness/ice drift of each season in the figures.
***Response:***
(1) The relative frequency (RF) is defined as following:

RF=$n$/N$_{grids}$, where $n$ represents the number of the grid cells in different thickness bins, and N$_{grids}$ is the sum of $n$ over all the bins.

(2) To better present the relative frequency of sea ice thickness and drift distribution in Fig. 8 and 9 in the previous manuscript *(now shown in Figure 8 and 9 in our manuscript with tracks)*, we used the heat maps to show the information instead, which are also shown in the following Figure 2 and 3. Instead of adding the mean ice thickness/ice drift of each season in the figures, we text it *in our manuscript with tracks in P9 line 263-265 and line 272-274* as:

➢ In statistics, the seasonal mean sea ice thickness are 2.06 m for spring, 2.11 m for summer, 1.32 m for autumn and 1.43 m for winter over the entire outlet, respectively.

➢ The seasonal mean sea ice velocity over the entire gate is larger than 5 km d$^{-1}$ except that is 3 km d$^{-1}$ in summer.

[Figure]

Figure 2. Seasonal variation of relative frequency (unit: %) of CMST sea ice thickness (unit: m) over the Fram Strait gate.

[Figure]

Figure 3. Seasonal variation of relative frequency (unit: %) of CMST sea ice drift (unit: km d⁻¹) over the entire Fram Strait gate.

**Point 7:** P7 L214-218: Could you explain why ice thickness is thickest in summer and ice drift is slowest in summer?

*Response:* We want to specifically explain that the seasonal mean sea ice thickness (effective thickness), drift and relative frequency are calculated along the Fram Strait gate defined in this study.

(1) The ULS observation (Hansen et al., 2013) showed that the amount of sea ice thicker than 5 m is in a large amount in May 1977 (30%), July 1998 (34%), June 2003 (30%) and February 2006 (31%). The maximum monthly mean ice thickness appearing in

May-August (summer months) covers 10 months of the total 17 months (Table 2 in Hansen et al., 2013). Then the monthly mean ice thickness maximum in CMST usually occurs in July (Figure 4 in blow, also shown in Figure 5 in the manuscript) could be reasonable. Because the thick ice advects from the area north of Greenland and CAA to the Fram Strait in summer, which can be easily observed from the daily spatial distribution of the thickness and also from the daily ice-age product from NASA (Arctic_Sea_Ice_Age_rev1.2160p30). However, this is not always the case, thus not all the summer ice is thicker than other months. The other reason can be rooted in the melt of the thin ice fraction. Figure 2 shows that, in summer, the 0-1 m fraction is almost gone due to melting, while it is well present in autumn and winter. Therefore, this will of course also affect the mean ice thickness distribution then.

In CMST, the seasonal sea ice thickness averaged over 6 years over ice covered area is 2.06 m for spring and 2.11m for summer. The summer ice is only slightly thicker than ice in spring in statistics. We realize that the expression in our previous manuscript was not proper and besides we find that the definition of different seasons in our study could also affect the robustness of this conclusion. We then refine this sentence to: "*Thin ice is more observed in Autumn and winter over the zonal gate according to the RF distribution in Figure 9. Although the maximum thickness over the entire Fram Strait occurs in May and June (Figure 5), higher RF in thick ice bins are found in summer (June, July and August in our definition) over zonal gate. Over the meridional gate, the ice thickness in summer and spring is almost uniformly distributed, while in August and Winter, high RFs are more found in thin ice bins. In statistics, the seasonal mean sea ice thickness are 2.06 m for spring, 2.11 m for summer, 1.32 m for autumn and 1.43 m for winter over the entire outlet, respectively.*"   (please see P9 line 259-269 in our manuscript with tracks)

(2) The sea ice velocity through the Fram Strait has high correlation with the cross-strait pressure gradient. The pressure gradient in summer is usually small, and thus results in slowest ice drift. This phenomenon has been investigated in previous studies (Kwok et al., 1999; Kwok et al, 2004; Vinje et al., 1998).

[Figure]

Figure 4. CMST sea ice thickness averaged over the entire Fram Strait gate, from September 2010 to December 2016. The black dotted line denotes monthly mean ice thickness based on CMST data with corresponding standard deviations while the blue dotted line represents monthly mean effective sea ice thickness of Ricker at al. (2018).

**Point 8:** P8 L224-225: better to mention this earlier in section 2.6.

*Response:* We mentioned this earlier in section 2.6 as suggested. *(please see P5 line 148-149)*

**Point 9:** Section 3.3 Sea ice volume export through the Fram Strait: It might be interesting to analyze the respective contributions of ice drift and ice thickness to the seasonal variation of sea ice export. How many percent of sea ice export variation can be explained by ice drift/ice thickness? More discussions could be made about the causes of the variations of ice drift, ice thickness and ice export during the melt season, e.g. what is the reason of the extreme low ice export in August 2015? Authors could also do some additional comparisons with atmospheric circulation variability, e.g. the Arctic Oscillation indices.

*Response:* Thank you for this suggestion. *(please see P10 line 299-316)*

(1) Following Ricker at al. (2018), we calculated the individual contribution of ice drift and ice thickness to the sea ice volume export. The correlation of determination ($R^2$) in the below Figure 5 *(also shown in Figure 11 in the manuscript with tracks)* shows that the sea ice drift variation contributions much more to the sea ice flux variation. However, the seasonal averaged drift and thickness are smoothed especially for sea ice drift variation. It is then not surprised that, in the seasonal time scale, we get nearly the same contribution from the sea ice drift and thickness with $R^2$ ranging between 0.36 and 0.46.

(2) The reason for the extremely low ice export in August 2015 is due to the rather smaller sea ice velocity (shown in Figure 5 below, *also shown in Figure 11 in our manuscript with tracks*).

(3) As suggested, we analyze the relations between atmospheric circulation (AO and NAO) and seasonal sea ice volume export through the Fram Strait (shown in Figure 6, also shown in Figure 12 in our manuscript with tracks). Results show that the correlation (r = 0.55) between AO index and ice volume flux is higher than r (Q, NAO) = 0.34. Both of our study and Ricker et al. (2018) find that AO may play a more important impact on the sea ice export (2011-2016) though the correlations in this study are smaller than that Ricker et al., (2018) since they are calculated during freezing season in their study.

[Figure]

Figure 5. Time series of standardized monthly mean sea ice volume export (red line) and corresponding monthly mean sea ice drift (blue line) and sea ice thickness (black line), including correlation of determination ($R^2$).

[Figure]

Figure 6. Time series of seasonal mean sea ice volume export (unit: km3, red line) and corresponding mean seasonal AO (blue line) and NAO (green line) index, including correlation coefficient (r).

**Point 10:** P8 L243: $1503\pm158$km$^3$ minus sign is missing. P10 L286: the values should have minus signs? P11 L315-316 and L319-320 are the same sentences.

*Response:* We checked through our manuscript and added the missing minus sign.

[revised manuscript text omitted]
 2011). The RF of sea ice thickness along the meridional gate also shows the major fractions appearing in the spring and summer. In addition, the values of seasonal mean sea ice thickness are 2.06 m for spring, 2.11m for summer, 1.32 m for autumn and 1.43 m for winter over the entire outlet, respectively.Figure 9approximately90),The seasonal mean sea ice velocity over the entire gate is larger than 5 km d$^{-1}$ except the 3 km d$^{-1}$ during summer.~~ And it can be found that the spring and winter ice concentration along the zonal gate is larger than that of summer and autumn.

**3.3 Sea ice volume export through the Fram Strait**

In this section, sea ice volume export over all seasons is investigated. Firstly, the examination of monthly Arctic sea ice volume export through the Fram Strait is demonstrated in Table 2. Both our results and Ricker et al. (2018) find that the maximum monthly sea ice export takes place in March 2011. The maximum of CMST data is -442 km$^3$  that is less than that (-540 km$^3$) of Ricker et al. (2018). Consistently, the lowest sea ice output for each study occurs in February 2011 when excluding the melt season (May-September). The minimum of the results shown in Ricker et al. (2018) is -21 km$^3$ while that is -34 km$^3$ in CMST data. Although

there are some differences in flux calculated based on CMST data and CryoSat-2 thickness and OSISAF drift data, both the estimations show a similar trend in annual cycle. Furthermore, the CMST data can provide sea ice variables (e.g., sea ice

285 thickness, concentration and drift) in the melt season that remote sensing retrieval data cannot cover. Taking advantage of CMST data, this study is trying to fill the research gap in the summer sea ice volume export. It is found that another minimum of ice export occurs in August 2015 because of the rather slow mean sea ice velocity (shown in Figure 11) during the study period. The minimum value for CMST is -11 km$^3$ that is 10 km$^3$ less than -21 km$^3$ (R) in February 2011 and 23 km$^3$ less than that for M2.

290 Moreover, the seasonal variation of sea ice export -though Fram Strait is shown in Figure 10. The ice volume output shows a significant seasonal variation. The seasonal maximums are found in spring of all years (2011-2016) and the low values usually occur in summer and autumn. The maximum seasonal ice export of -970 (±60) km$^3$ (sea ice volume export has been rounded off to significant figures in seasonal and interannual time scales) takes place in the spring of 2012 owing to both simultaneously faster ice drift and thickness, while the minimum flux of

295 -240 (±40) km$^3$ occurs in autumn of 2016 caused by simultaneously rather slower ice motion and thickness. Unlike other autumn ice export, the ice volume export of autumn 2013 abnormally increases and reaches -620 (±60) km$^3$. This abnormal increase can be also explained by the faster ice drift (shown in Figure 9).

Furthermore, we standardize the sea ice volume export, ice drift and thickness and then calculate the correlations of

300 determination (R$^2$) between monthly sea ice volume export and thickness, and also for drift (shown in Figure 11). R$^2$ between monthly mean sea ice flux and drift is 0.77, which is much higher than R$^2$(Q$_{EX}$, 
[revised manuscript text omitted]

---

## Author Response (AR1)

**Response**

Dear Editor:

We would like to thank Prof. Julienne Stroeve for the time and efforts on the review. We further revised the manuscript by rephrasing sentences, rearrange the paragraphs and grammar checking without changing the meaning in the previous version. The manuscript with mark-ups is enclosed below.

Thanks for your considerations for the possible publication on TC.

Best regards,
Longjiang Mu
On behalf of co-authors
Email: longjiang.mu@awi.de

[revised manuscript text omitted]

---

## Author Response (AR2)

**Responses**

Dear Editor,

We thank the Editors and Reviewers for their scientific suggestions during the review process. The final manuscript and figures are submitted for the production process.

Best regards,
Longjiang Mu
on behalf of all co-authors.